# Single-cell transcriptomic atlas of lung microvascular regeneration after targeted endothelial cell ablation

**Rafael Soares Godoy**[1,2†], **Nicholas D Cober**[1,2,3†], **David P Cook**[1,3], **Emma McCourt**[1], **Yupu Deng**[1,2], **Liyuan Wang**[1,2,3], **Kenny Schlosser**[1,2], **Katelynn Rowe**[1,2], **Duncan J Stewart**[1,2,3]*

[1]Ottawa Hospital Research Institute, Ottawa, Canada; [2]Sinclair Centre for Regenerative Medicine, Ottawa, Canada; [3]Department of Cellular and Molecular Medicine, University of Ottawa, Ottawa, Canada

**Abstract** We sought to define the mechanism underlying lung microvascular regeneration in a model of severe acute lung injury (ALI) induced by selective lung endothelial cell ablation. Intratracheal instillation of DT in transgenic mice expressing human diphtheria toxin (DT) receptor targeted to ECs resulted in ablation of >70% of lung ECs, producing severe ALI with near complete resolution by 7 days. Using single-cell RNA sequencing, eight distinct endothelial clusters were resolved, including alveolar aerocytes (aCap) ECs expressing apelin at baseline and general capillary (gCap) ECs expressing the apelin receptor. At 3 days post-injury, a novel gCap EC population emerged characterized by de novo expression of apelin, together with the stem cell marker, protein C receptor. These stem-like cells transitioned at 5 days to proliferative endothelial progenitor-like cells, expressing apelin receptor together with the pro-proliferative transcription factor, *Foxm1*, and were responsible for the rapid replenishment of all depleted EC populations by 7 days post-injury. Treatment with an apelin receptor antagonist prevented ALI resolution and resulted in excessive mortality, consistent with a central role for apelin signaling in EC regeneration and microvascular repair. The lung has a remarkable capacity for microvasculature EC regeneration which is orchestrated by newly emergent apelin-expressing gCap endothelial stem-like cells that give rise to highly proliferative, apelin receptor-positive endothelial progenitors responsible for the regeneration of the lung microvasculature.

**\*For correspondence:**
djstewart@ohri.ca

†These authors contributed equally to this work

**Competing interest:** The authors declare that no competing interests exist.

## Editor's evaluation

The manuscript by Godoy and colleagues is an important contribution to the understanding of how lung endothelial regeneration progresses following endothelial ablation. The novelty and elegance of this study are rooted in the regional and specific ablation of lung endothelial cells using diphtheria toxin without the massive inflammatory activation that is seen with lung injury induced by bacterial infections, viral infections, or lipopolysaccharide. The data convincingly demonstrate that there is an emergence of a highly proliferative lung endothelial subpopulation with specific molecular signatures that facilitate regeneration.

## Introduction

ALI and its severe clinical counterpart, the adult respiratory distress syndrome (ARDS), remains a major cause of morbidity and mortality in critically ill patients, accounting for ~30% of ICU admission with 28 day mortality approaching 40% (*Ranieri et al., 2012*; *Wheeler and Bernard, 2007*).

Despite decades of research, no specific therapies have been developed that improve outcomes in ARDS (*Ranieri et al., 2012*). The COVID-19 pandemic has highlighted the devastating nature of this condition as infection with the SARS CoV2 virus leads to a particularly severe form of ARDS (*Fan et al., 2020*; *Batah and Fabro, 2021*) which has been responsible for the vast majority of over 5 million deaths world-wide. COVID-19-associated ARDS has a strong vascular component which is characterized by intense endothelial inflammation (endothelialitis) and necrosis (*Ackermann et al., 2020*; *Lowenstein and Solomon, 2020*; *Joffre et al., 2022*), consistent with the emerging role of endothelial injury in other forms of ALI leading to the breakdown of the air-blood barrier (*Jin et al., 2020*). Indeed, recent reports have suggested that endothelial repair is required for the resolution of ALI (*Minamino and Komuro, 2006*; *Huang et al., 2012*), and thus an important target for the development of novel therapeutic strategies. Unfortunately, little is known about the mechanisms that underlie lung microvascular repair and its role in ALI resolution. In experimental models the transcription factor, Forkhead box M1 (*Foxm1*), has been implicated as a driver of endothelial cell (EC) proliferation and microvascular repair that is required for ALI resolution (*Huang et al., 2012*; *Zhao et al., 2014*; *Mirza et al., 2010*), and apelin has been reported to protect against inflammation and oxidative stress (*Zhang et al., 2018*). During angiogenesis, apelin is induced in endothelial tip cells by tissue hypoxia and VEGF and signals to trailing stalk ECs that express the apelin receptor (*Eyries et al., 2008*; *Kidoya and Takakura, 2012*). Even though apelin is known to play an important role in vascular development and angiogenesis (*Kidoya and Takakura, 2012*; *Yan et al., 2020*), the relevance of this peptide for microvascular repair in ALI and ARDS has not been explored.

It has increasingly been recognized that ECs play a key role in orchestrating tissue repair through the self-renewal and differentiation of resident stem and progenitor cells in an organ-specific manner (*Rafii et al., 2016*). The understanding of this heterogeneous EC landscape has been greatly facilitated by the introduction of single-cell transcriptomics analysis. Using this approach, two specialized lungs microvascular EC populations have recently been described in the normal lung *Gillich et al., 2020*; 'aerocytes' (aCap ECs), which are characterized by the expression of apelin, and general (gCap) ECs expressing the apelin receptor. Aerocytes are highly differentiated, large cells that make up the endothelial component of the alveolar air-blood barrier and are incapable of proliferation (*Gillich et al., 2020*). In contrast, gCap ECs are smaller and located at thicker regions of the alveolar wall (*Gillich et al., 2020*) and respond to injury by proliferation; therefore, represent the lung EC population within which endothelial stem cells may arise.

To better understand the role of endothelial injury and repair in ALI, we established a new model induced by targeted lung EC ablation. We now demonstrate that EC ablation was sufficient to result in severe ALI, with all the features of standard models induced by inflammatory or toxic agents. Remarkably, animals survived the loss of up to 70% of the lung vascular endothelium by virtue of rapid EC regeneration initiated by the emergence of a new gCap endothelial stem cell population post-injury that paradoxically exhibited the de novo expression of apelin together with a stem cell marker, protein C receptor (*Procr*). This population rapidly transitioned to highly proliferative progenitor-like cells, characterized by apelin receptor and *Foxm1* expression, which were responsible for repopulating all depleted lung endothelial fields, including aerocytes, by an apelin-dependent mechanism leading to rapid ALI resolution.

## Results

### Selective lung EC ablation in Cdh5-DTR mice

Binary transgenic animals harboring Cdh5-cre-iDTR were obtained by crossing homozygous DTR and Cdh5 mice to generate double heterozygous offspring (*Figure 1A*). Administration of DT at doses below 20 ng IT was consistent with survival (*Figure 1—figure supplement 1A*) resulting in modest increases in right ventricular systolic pressure (RVSP) that was maximal at 10 ng of DT (*Figure 1—figure supplement 1B*). DTR expression was localized to ECs in Cdh5-DTR animals (*Figure 1B*) and widespread apoptosis was seen at 3 days post DT administration (10 ng) in Cdh5-DTR mice (*Figure 1C*). Indeed, DT administration resulted in a ~70% reduction in EC numbers at 3 days by flow cytometry (p<0.001) (*Figure 1D and E*) with full recovery by day 7 (*Figure 1E*). Lung permeability was increased (p<0.01) at 3 days post DT administration (*Figure 2A and B*), again returning to normal by 7 days, with no evidence of increased vascular permeability in other organs (*Figure 2B* and *Figure 1—figure*

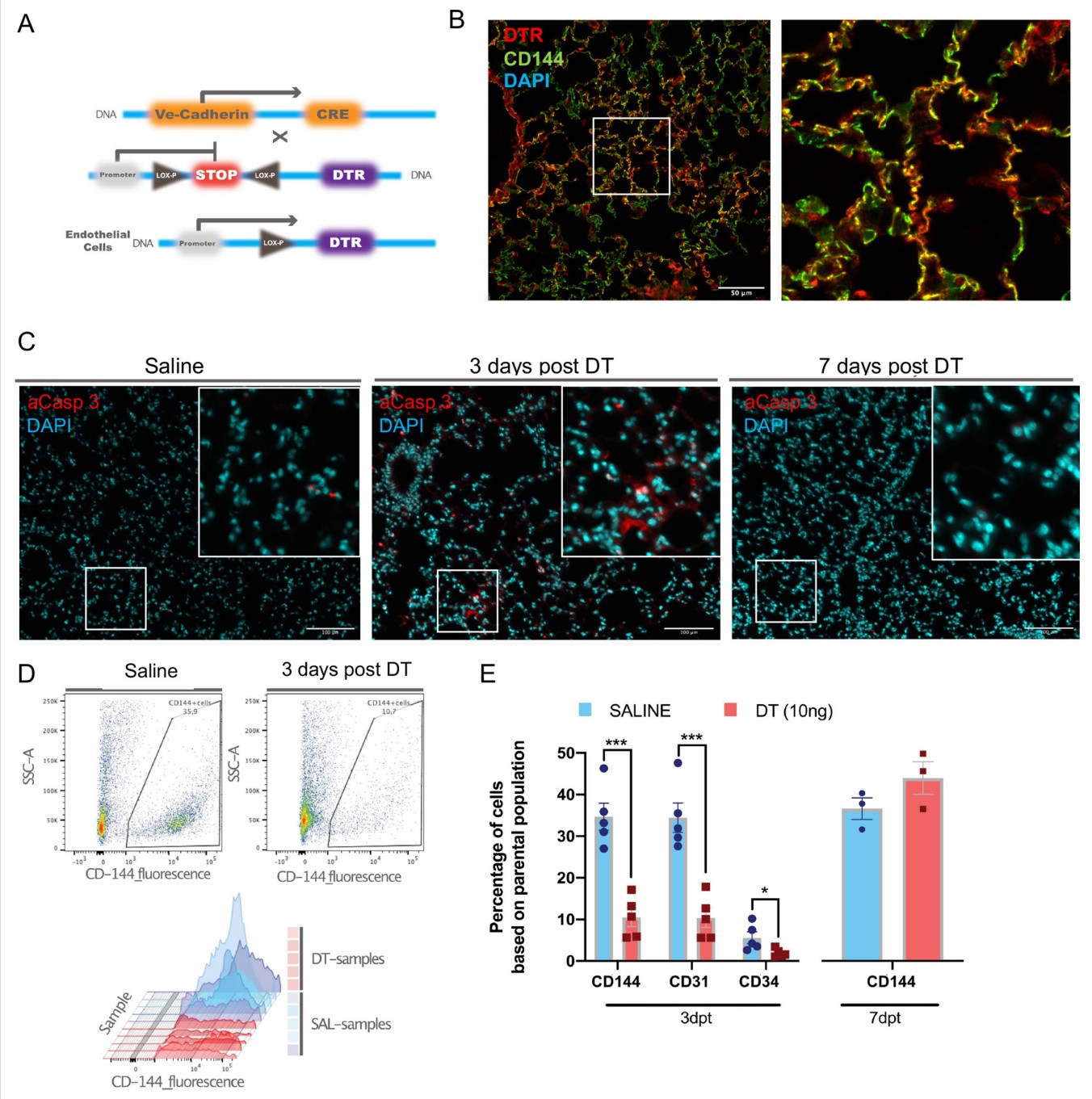

**Figure 1.** Establishment of the diphtheria toxin (DT)-induced endothelial cell (EC) ablation model. (**A**) Transgenic mice harboring Cre recombinase cDNA downstream from a 2.5 kb fragment of the VE-Cadherin (Cdh5) mouse promoter (B6.FVB-Tg(Cdhn5-cre)7Milia) were crossed with mice with Cre-inducible expression of DTR (C57BL/6-Gt(ROSA)26Sor$^{tm1(HBEGF)Awai}$/J) giving rise to Cdh5-DTR binary transgenic mice. (**B**) Immunostaining for human diphtheria toxin receptor (DTR) (red) and ECs (anti-CD144, green) and merged image showing co-localization (yellow) in lung sections from Cdh5-DTR binary transgenic mice 3 days post intra-tracheal (IT) delivery of saline or DT (10 ng). Scale bar is 50μm. (**C**) Immunostaining for activated caspase 3 (aCasp-3, red) in lung sections from binary transgenic mice 3 or 7 days after treatment DT or saline (DAPI nuclear staining in blue). Scale bar is 100μm. (**D**) Representative plots of lung EC numbers assessed by flow cytometry (CD144) in binary transgenic mice 3 days after IT delivery of saline or DT. (**E**) Summary flow cytometric data showing the percent of total lung cells staining positive using CD144, CD31, or CD34 antibodies in DT-treated binary transgenic mice 3 or 7 days post-treatment (dpt) compared with saline. Data represented as mean ± SEM. n=5 for 3d timepoint, n=3 for 7d timepoint, multiple unpaired *t*-tests were performed with multiple comparisons using Holm-Sidak correction. More details are in *Figure 1—figure supplement 1*.

The online version of this article includes the following figure supplement(s) for figure 1:

**Figure supplement 1.** Sublethal dose of diphtheria toxin (DT) induces pulmonary hypertension and selective increase in lung permeability.

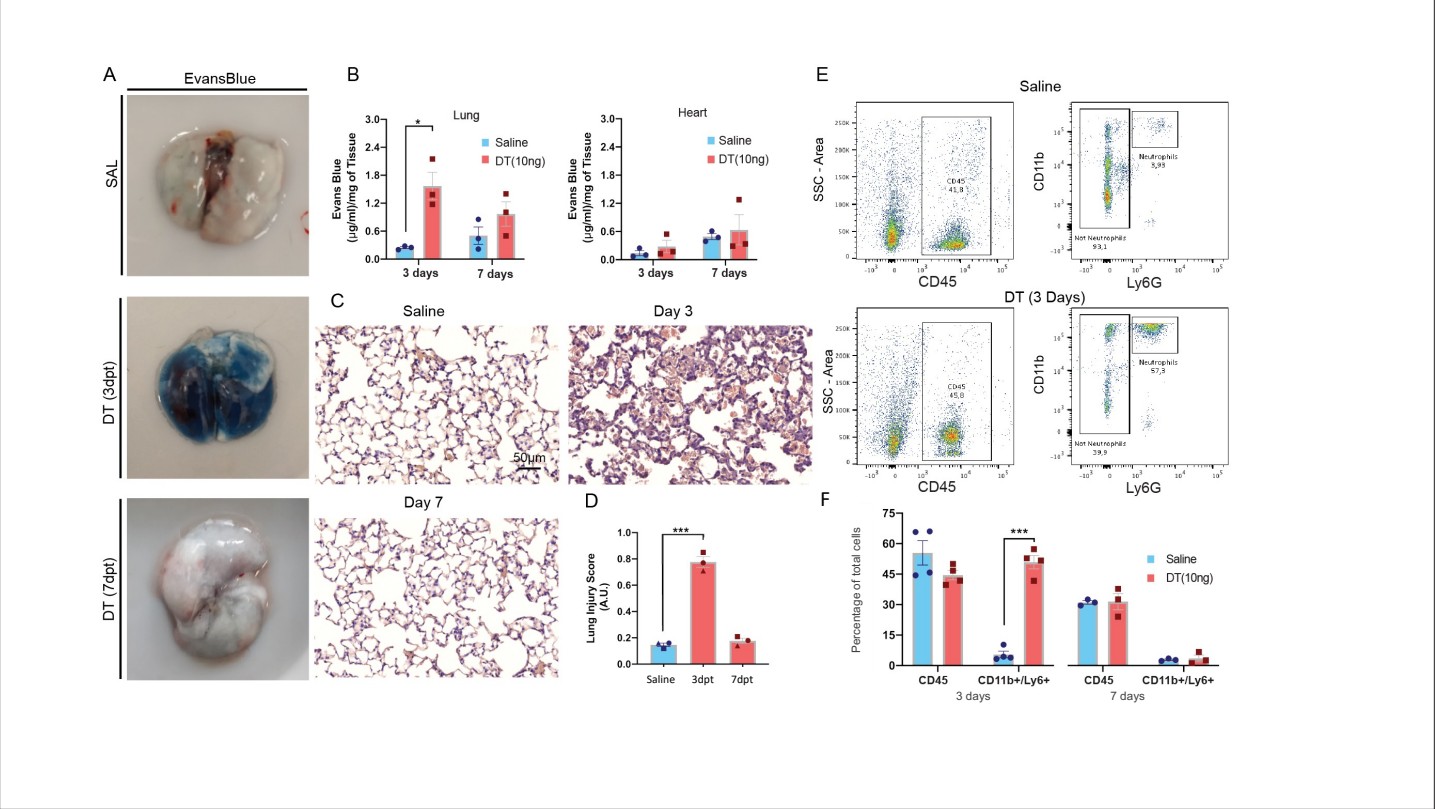

**Figure 2.** Diphtheria toxin (DT) administration results in an acute lung injury phenotype. (**A**) Representative examples of Evans blue staining of lungs from binary transgenic mice. (**B**) Summary data showing a marked increase in Evans Blue lung content in binary transgenic mice treated with DT (10 ng) or saline at 3 and 7 days post-treatment (dpt). (**C**) Representative histological lung sections (H&E staining) from binary transgenic mice at 3 and 7 days post intra-tracheal (IT) administration of DT or saline with the corresponding summary data (**D**) for a validated lung injury score. (**E**) Representative examples of flow cytometry plots showing gating strategy for assessing CD11b/Ly6G positive leukocytes in lungs from binary transgenic mice 3 days after IT delivery of DT or saline with summary data (**F**) at 3 and 7 days post-treatment (dpt). Data represented as mean ± SEM. An unpaired multiple *t*-tests with Holm-Sidak multiple comparisons method with alpha (0.05) was used for the analysis of data presented in panels B and F, whereas one-way ANOVA corrected for multiple comparisons with Dunnett was conducted for panels D. n=3-4 biological replicates per group, *=p<0.05; **=p<0.01; ***=p<0.005. Scale bar is 50μm in panel C.

supplement 1C). At 3 days, there was a marked increase in the lung injury score (*Figure 2C and D*) consistent with severe ALI, and this was associated with the influx of CD11b+Ly6G+ neutrophils by flow cytometry (*Figure 2E*).

## Changes in global lung cell populations with single-cell transcriptomic profiling

Multiplexed scRNA-seq analysis was performed on lung tissues of Cdh5-DTR mice at baseline (day 0) and 3, 5, and 7 days post DT administration (*Figure 3A*). Cells lacking a barcode, or positive for multiple barcodes were excluded from further analysis (*Figure 3—figure supplement 1*). Uniform Manifold Approximation and Projection (UMAP) dimensionality reduction maps of the data revealed 35 separate cell populations, representing immune, endothelial, stromal, and epithelial cells (*Figure 3B* and *Figure 3—figure supplement 2*). Endothelial and monocyte-macrophage populations showed the greatest changes in gene expression profiles after DT administration (*Figure 3C*). Cell types that were most affected by DT treatment were identified using Augur (*Skinnider et al., 2021*) and included ECs (Clusters 4, 5, and 0) and macrophages (Clusters 20 and 21) (*Figure 3D*). Pericytes and type 2 pneumocytes (Clusters 31 and 8) were also among the top 10 most affected by EC ablation.

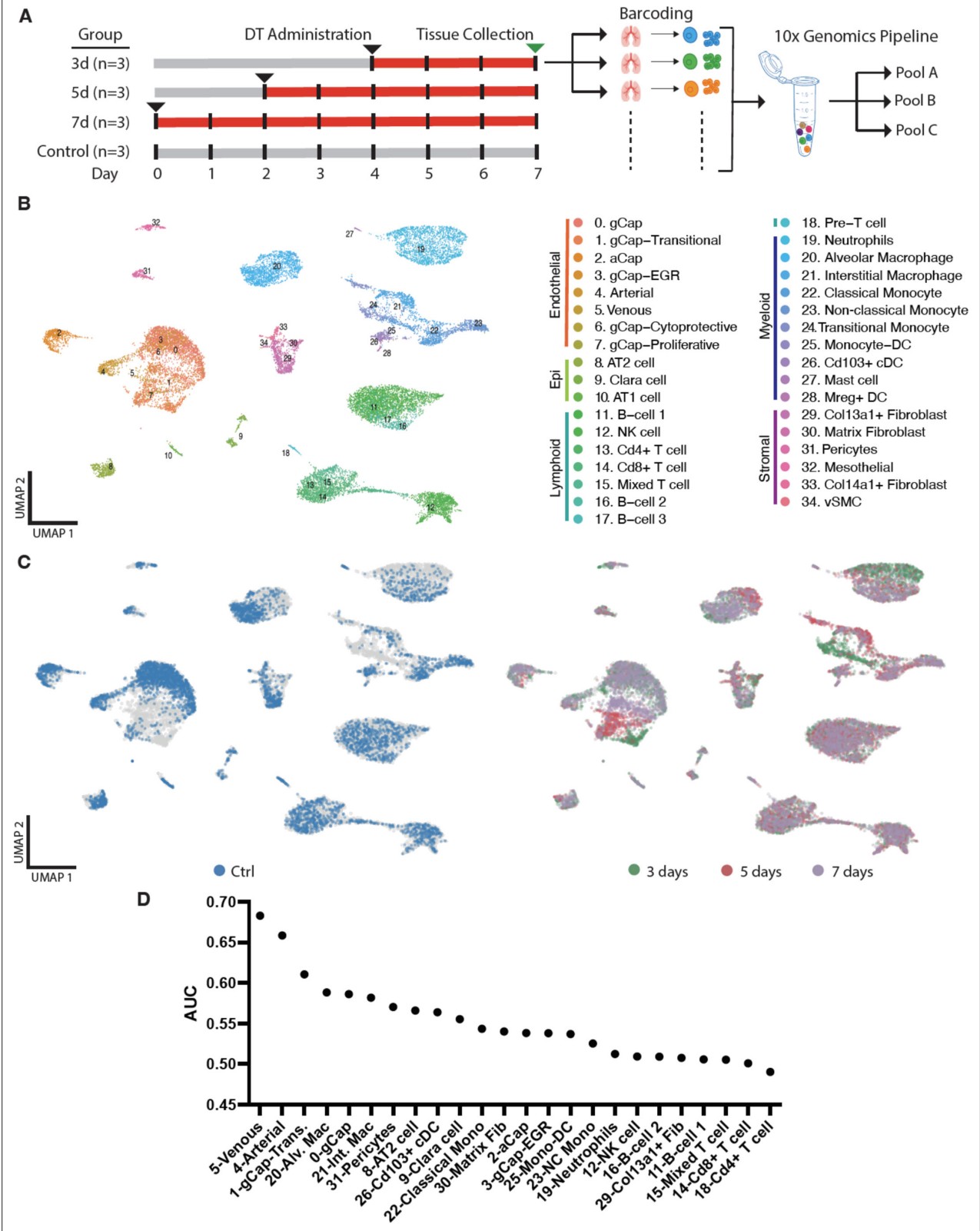

**Figure 3.** Multiplexed single-cell transcriptomic analysis after diphtheria toxin (DT)-induced lung endothelial cell (EC) ablation. (**A**) Schematic of workflow showing the experimental design. Three separate cohorts (3, 5, and 7 days) of binary transgenic mice received DT (IT, 10 ng), and a fourth cohort of healthy animals served as a control, with the timing of DT delivery such that all mice were sacrificed on the same day. Three animals (biological replicates) were included per group. Lungs cells were immediately isolated and barcoded to identify individual donor animals, then pooled

*Figure 3 continued on next page*

*Figure 3 continued*

and subjected to library construction using 10x-Genomics Single-cell 3' RNA sequencing kit v.3. Global plot of all lung cells at all time points using uniform manifold approximation and projection (UMAP). (**B**) 35 distinct populations were identified and could be assigned into five major categories: endothelial, epithelial stromal, myeloid, and lymphoid. (**C**) UMAP plots of global lung cells are shown for each mouse cohort: control (blue); day 3 (green), day 5 (red), and day 7 (purple) (**C**). (**D**) A machine learning model was used to predict cells that become more separable during treatment based on their molecular measurements (*Skinnider and Lin, 2021*; *Skinnider et al., 2021*). More details for this analysis are provided in *Figure 3—figure supplements 1–2*.

The online version of this article includes the following figure supplement(s) for figure 3:

**Figure supplement 1.** Quality control for single-cell RNA sequencing (scRNA-seq).

**Figure supplement 2.** Major cell type classification.

## Changes in EC populations in response to injury

We identified eight distinct ECs Clusters (*Figure 4A*), including five populations that exhibited gene expression profiles typical for alveolar gCap ECs (Clusters 0, 1, 3, 6, and 7) and one that corresponded to aCap ECs (Cluster 2), which are also called aerocytes (*Figure 4—figure supplement 1*; *Gillich et al., 2020*; *Niethamer et al., 2020*). Cluster 1 was unique in that it was made up of 4 distinct 'zones'; each zone specific to a single time point before and after EC ablation (*Figure 4B*), whereas Cluster 7 emerged mainly at 5 days after EC injury and was characterized by a high expression of *Mki67*, a marker of proliferating cells (*Figure 4C*). Most endothelial populations, including aCap ECs, exhibited a marked decline in cell numbers after DT-induced EC ablation, some reaching a nadir of almost 90% cell loss at 5 days (*Figure 4D*). Only Clusters 1 and 7 showed stable or increased cell numbers, and these were the only clusters that showed evidence of EC proliferation, consistent with their role in lung microvascular repair.

To explore the role of ECs in Cluster 1 in regenerating microvascular ECs after endothelial injury, we performed RNA velocity and trajectory analysis using scVelo (Python) (*Bergen et al., 2020*). This revealed strong vectors from Cluster 1 to other EC Clusters that were depleted by EC ablation, including Clusters 0 and 3 (gCap ECs), Cluster 4 (venous), and Cluster 5 (arterial) (*Figure 5A*), consistent with a central role of this transitional cluster in microvascular regeneration and repair. This was also supported by evidence of DNA synthesis (S score), localized to Zone 2, and cell proliferation (G2M score) within Zone 3 of Cluster 1 as well as the proliferative Cluster 7 (*Figure 5B*). Moreover, we identified the top 15 driver genes implicated in these transitions (*Figure 5C*), with eight genes associated with the transition from Cluster 1 to Cluster 0, 5 genes involved in the transition of Cluster 1 to aCap ECs (Cluster 2), and 2 genes for Cluster 1 to venous ECs. Finally, an inferred latent time plot of the 300 top driver genes revealed the same temporal progression of gene expression as observed with the independent timepoint analysis (*Figure 5D*). This originated with Cluster 1, progressed through Cluster 0, and terminated in Cluster 2 at the end of the pseudotime cascade. Therefore, despite the limitation that the velocity analysis only infers a timeline based on genomic cues (i.e. the proportion of spliced RNA), and is agnostic to the actual timing of the cell harvesting in the serial transcriptomic dataset, it approximates remarkably well the temporal shifts in EC populations that were demonstrated by serial scRNA-seq at defined timepoints.

## Temporal sequence of angiogenic gene expression in aCap and gCap ECs

At baseline (day 0), apelin was found to be uniquely expressed in aCap ECs of Cluster 2 (*Figure 6A*) as previously reported (*Gillich et al., 2020*; *Vila Ellis et al., 2020*), whereas apelin receptor (*Aplnr*) was found only in gCap ECs. In addition to apelin, aCap ECs also exhibited predominant expression of other EC tip cell genes including *Kdr (Vegfr2)*, *Npl1* (neuropilin 1), and *Cd34* (*Figure 6—figure supplement 1A*; *Blanco and Gerhardt, 2013*; *Zecchin et al., 2017*), whereas stalk cell markers, such as *Hey1, Flit1, and Notch1* (*Figure 6—figure supplement 1B*) was mainly expressed by gCap ECs. Three days after EC injury, de novo expression of apelin was seen in the newly emergent gCap EC population in Zone 2 of Cluster 1 (*Figure 6A*). This population robustly expressed gCap markers, with the exception of *Aplnr* (*Figure 4* and *Figure 4—figure supplement 1*). Interestingly, they also exhibited the expression of *Procr* (protein C receptor or EPCR; *Figure 6A*), a marker of bipotent resident vascular endothelial stem cells (*Yu et al., 2016*). Co-expression of apelin with *Procr,* as well as *Cd93*

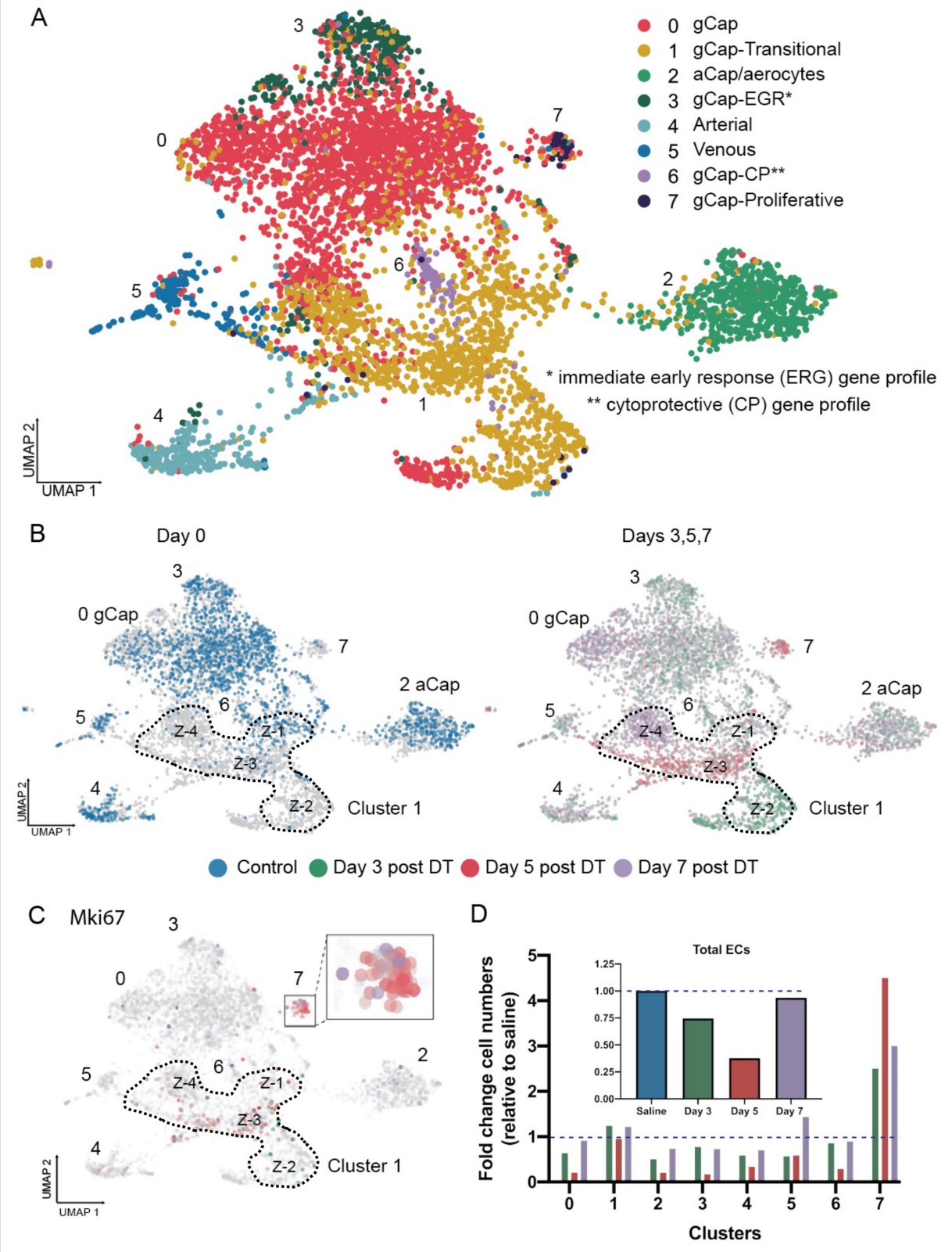

**Figure 4.** Endothelial cell populations in control and diphtheria toxin (DT)-treated binary transgenic diphtheria toxin receptor (DTR) mice. (**A**) Uniform manifold approximation and projection (UMAP) representation of the re-clustered endothelial populations for all cohorts produced eight distinct endothelial cell (EC) clusters. (**B**) The UMAP plots for each cohort are shown separately: control (blue); day 3 (green), day 5 (red), and day 7 (purple) The dotted line delineates Cluster 1 which was unique in that it consisted of four zones (Z-1 to Z-4) each specific to a single time cohort. (**C**) Cell proliferation,

*Figure 4 continued on next page*

*Figure 4 continued*

identified by *Mki67* expression, was seen only at day 5 (red) in Zone 3 of Cluster 1 and Cluster 7. (**D**) Change in numbers of lung ECs in the three cohorts treated with DT (days 3, 5, and 7) are expressed as fold-change relative to control for all EC populations (insert) and numbers of ECs broken down for each EC cluster. Typical aerocytes (aCap) and general capillary (gCap) gene expression profiles are shown in *Figure 4—figure supplement 1*.

The online version of this article includes the following figure supplement(s) for figure 4:

**Figure supplement 1.** Aerocytes (aCap) and general capillary (gCap) genes in endothelial cells (EC) populations pre and post-injury.

(a gCap marker) was evident at 3 days post-injury (*Figure 6B*) and confirmed by immunofluorescence staining (*Figure 6C*). To our knowledge, this represents the first demonstration of apelin expression by gCap ECs. As well, *Angpt2* (angiopoietin 2) (*Maisonpierre et al., 1997*) was uniquely expressed in this endothelial stem cell-like population (*Maisonpierre et al., 1997*), together with the progenitor cell marker, *Cd34* (*Figure 6—figure supplement 1A*).

At day 5 post-DT, new populations emerged in Zone 3 of Cluster 1, as well as in Cluster 7 (*Figure 4B*) exhibiting expression of *Mki67*, a marker of cell proliferation, together with the *Aplnr* (*Figure 6A*) and *Foxm1*, a pro-proliferative transcription factor (*Figure 6—figure supplement 1B*); however, the density of proliferating cells was greatest in Cluster 7. By day 7, the expression of PROCR/EPCR was nearly entirely lost (*Figure 6C*). This was consistent with a restoration of the baseline profile of apelin expression being restricted to aCap ECs (aerocytes).

Next, we performed an unbiased analysis of the role of other transcription factors using the decoupleR software package which infers TF activity based on known TF-gene set interactions (*Badia-I-Mompel et al., 2022*). This revealed 25 TFs that were significantly up or downregulated in our dataset, with very distinct profiles in the different ECs populations (*Figure 6—figure supplement 2*). Not surprisingly, the greatest increase in TF activity was seen in the proliferative Cluster 7, with *Foxm1* being among the top candidates. Other TFs known to be involved in cell cycle regulation were also identified, including three members of the *E2f* family, *Myc*, and *Tfdp1*. As well, strong *Nanog* and *Sox10* activity, which play important roles in stem cells and developmental biology, was unique to aCap ECs. In Cluster 3, which is characterized by high expression of early response genes such as *Fos*, we saw the strong activity of the Ets family member, *Elk4*, which binds to the serum response element in the promoter of the *Fos*.

To determine whether the same regenerative EC populations could also be identified in commonly used ALI models, we accessed a publicly available single-cell transcriptomics database (GSE148499) from a recently published model of ALI induced by intraperitoneal injection of LPS (*Zhang et al., 2022*). In this study, scRNA-seq was performed on lung ECs at 6 timepoints after LPS administration (0, 0.25, 1, 2, 3, and 7 days) with a separate analysis at each timepoint. We reanalyzed these data by combining all timepoints to evaluate the transition of EC populations in this model of ALI. This revealed a number of distinct gCap EC clusters, each representing a different time point (*Figure 6—figure supplement 3A–C*), very similar to the zones of the transitional Cluster 1 of our dataset. At baseline (day 0), apelin was only expressed by the aCap ECs, and *Procr* was not expressed at all (*Figure 6—figure supplement 3D*). After LPS-induced injury, a gCap EC cluster co-expressing *Apln/Procr* was apparent as early as 6 hr post-LPS, followed by the emergence of a proliferative *Mki67/Foxm1*-postive population at day 3. These results suggest that the novel regenerative endothelial populations that we identified after EC ablation are relevant to commonly used models of ALI.

## Differential gene expression and pathway enrichment analysis of endothelial clusters

*Figure 7A* shows volcano plots of differentially expressed genes (DEGs) in EC clusters compared to Cluster 0 and the top 10 DEGs are presented in *Figure 7B*. For this analysis, only Zone 1 of Cluster 1 was included, representing day 0 (baseline). Not surprisingly, Cluster 1 exhibited a high degree of similarity in gene expression profile with Cluster 0, the largest gCap population, although pathway enrichment analysis showed an increase in activation of pathways associated with oxidative phosphorylation, as well as EIF2 signaling which was also shared by Clusters 4 and 5 (arterial and venous ECs, respectively; *Figure 7C*). As expected, Cluster 2 (aerocytes) showed high expression of typical aCap genes, such as *Endrb*, *Fibin*, and *Car4*, and enrichment in pathways related to PKA and Ephrin B receptor signaling (*Figure 7B and C*). Interestingly, Cluster 3 showed upregulation in many early

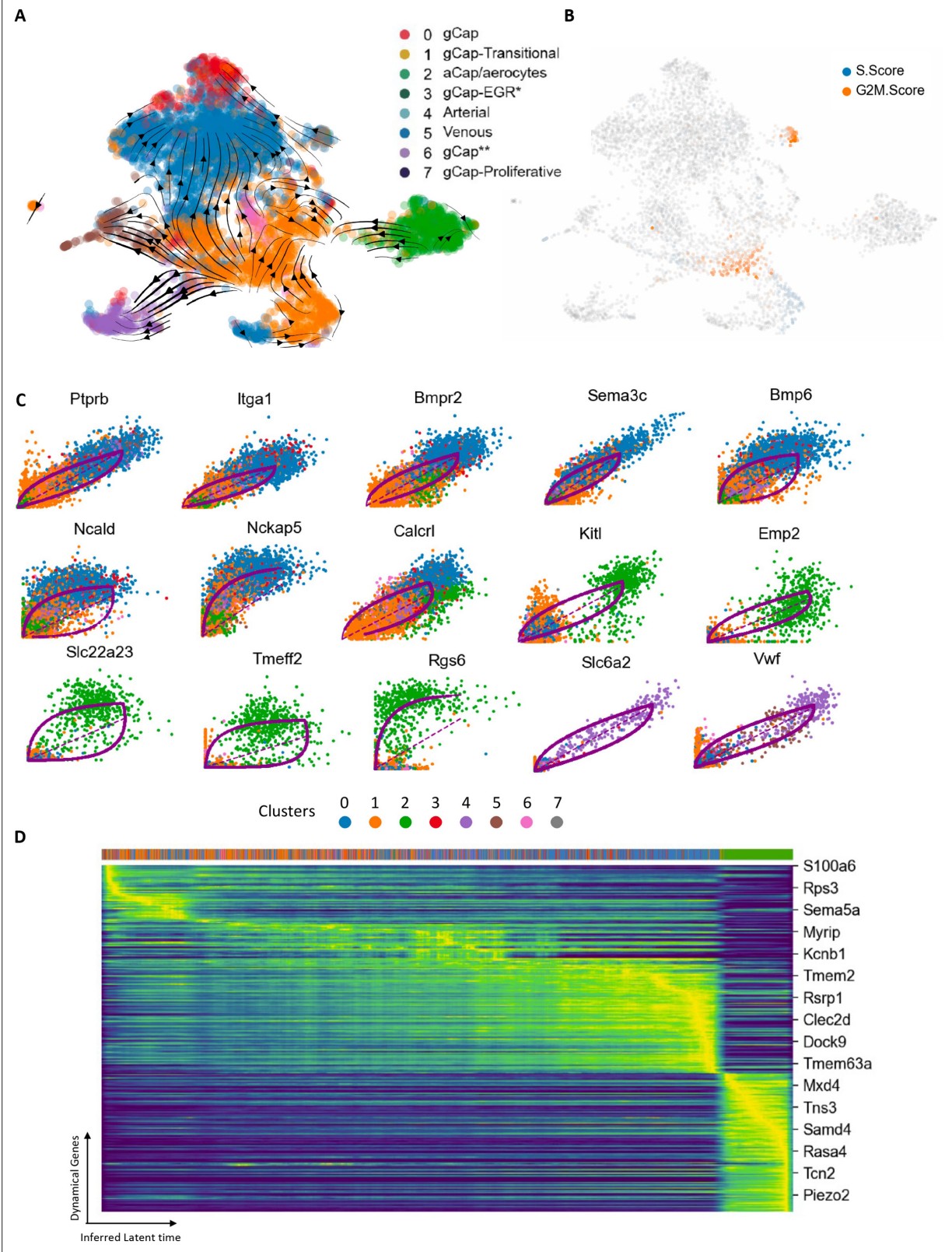

**Figure 5.** RNA velocity and trajectory analysis. (**A**) This model showed a dynamic shift of endothelial cells (ECs) through Zones 2 and 3 of transitional Cluster 1 with strong vectors towards general capillary (gCap), arterial, and venous EC populations. (**B**) Zone 2 ECs of Cluster 1 exhibited evidence of DNA synthesis (blue), whereas cells in Zone 3, and proliferative cluster 7, showed increased G2M scores, consistent with their role in repopulating the lung microvasculature ECs. (**C**) Top 15 genes involved in specific transitions between clusters: eight genes implicated in the transition from Cluster 1 to

*Figure 5 continued on next page*

*Figure 5 continued*

Cluster 0; five genes for Cluster 1 to aCap ECs (Cluster 2), and two genes for Cluster 1 to venous ECs (Cluster 4). (**D**) Cascade of 300 top genes in the dynamical gene expression analysis. Horizontal ribbon on top identifies clusters by color coding according to the legend in panel A. Genes listed on the right was randomly selected. There was a clear progression of gene expression beginning with Cluster 1 (orange) and progressing through to Cluster 0 (blue) and finally Cluster 2 (green) at the end of the pseudotime cascade.

response genes, including *Erg1*, *Junb*, *Ier2*, *Ier3*, and *Fosb* and genes associated with enrichment in a number of inflammatory signaling pathways shown in *Figure 7C* including IL17A, iNOS, Toll-like receptor, and Il-6; whereas Cluster 6 exhibited no significant predicted activation or inhibition in any pathways. EC Cluster 7, exhibited the greatest number of DEGs (~1200), nearly all of which were upregulated, including many genes involved in cell proliferation such as *Rrm2*, *Aurkb*, *Cdc25c*, *Tk1*, *Cdca8* and *Birc5*, and unique activation of signaling pathways involved in cell cycle regulation (*Figure 7B and C*). The identities of all significantly enriched canonical pathways are provided in *Figure 7—figure supplement 1*.

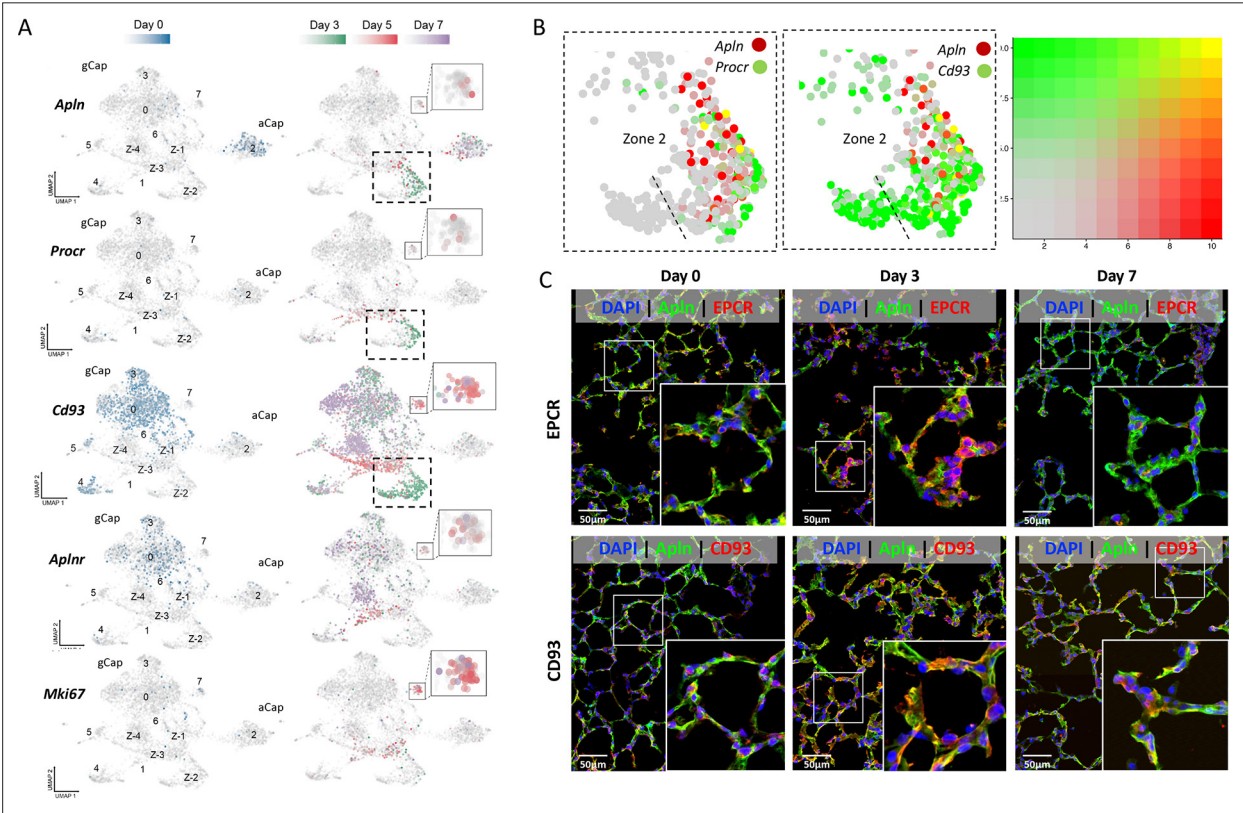

**Figure 6.** Apelin expression in a transitional general capillary (gCap) stem-like endothelial cell (EC) population at 3 days post-injury. (**A**) Under basal conditions (day 0; blue color) *Apln* was only expressed by aCap ECs (aerocytes; Cluster 2), whereas *Aplnr* expression was seen in gCap ECs (*Cd93* positive). 3 days after EC ablation (Green color), de novo apelin expression was apparent in gCap ECs of Zone 2, Cluster 1, together with *Procr*. At 5 days post-EC injury (Red color), *Aplnr* expression was seen in the adjacent Zone 3 of Cluster 1 and in Cluster 7. (**B**) Two-gene analysis of Zone 2 of Cluster 1 showing co-expression of *Apln* with *Procr* (left panel) and *Cd93* (middle) at 3 days post-injury. Right panel shows 2-color scale with yellow indicating a complete overlap in expression and faded red/green partial overlap. (**C**) Immunofluorescence staining showing colocalization (yellow) of APLN, EPCR, and CD93 in Zone 2 of Cluster 1 only at 3 days post EC ablation. Scale bar is 50μm. More details in *Figure 6—figure supplements 1–3*.

The online version of this article includes the following figure supplement(s) for figure 6:

**Figure supplement 1.** Temporal evolution of expression of tip and stalk cell genes pre and post-endothelial cell (EC) ablation.

**Figure supplement 2.** Unbiased inference of transcription factor (TF) activity based on known TF-gene set interactions (decoupleR) showing the top 25 significantly up- or down-regulated TFs.

**Figure supplement 3.** Serial single-cell transcriptomic analysis of a publicly available dataset endothelial cell (EC) gene expression in an endotoxin (LPS) model of acute lung injury (DOI: 10.1172/jci.insight.158079).

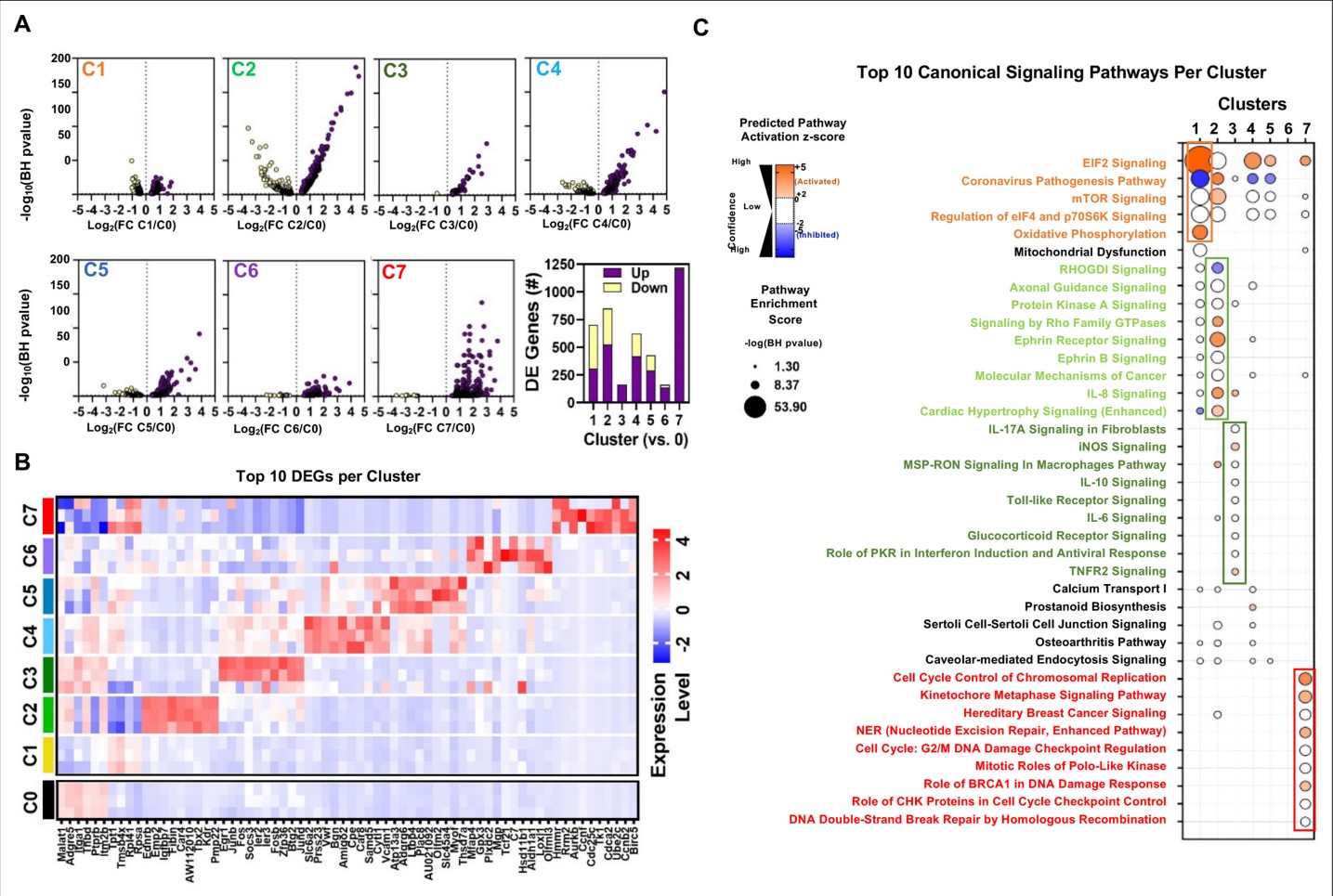

**Figure 7.** Differential gene expression and pathway enrichment analysis of endothelial sub-Clusters 1–7. (**A**) Volcano plots showing the distribution of differentially expressed genes in Clusters 1–7 (C1 to C7) according to statistical significance (Benjamini-Hochberg (BH) adjusted p-value), magnitude and direction of fold change (FC) relative to Cluster 0 (at baseline time 0). Only genes that passed the false discovery rate threshold (FDR) <0.05 are shown. Bar chart summarizes the number of differentially expressed (DE) genes in each cluster (versus Cluster 0). Purple and yellow bars are stacked to show the proportion of genes that were upregulated and downregulated, respectively. (**B**) Heatmap shows expression levels of 68 genes representing the combined top 10 differentially expressed genes in Clusters 1–7 (versus Cluster 0) at baseline day 0. Mean expression levels (z-score of log-transformed normalized counts) are shown for n=3 mice/cluster. (**C**) Pathway enrichment analysis using DE gene sets from panel A with Ingenuity Pathway Analysis. Identity of 38 canonical signaling pathways representing the combined top 10 most significantly enriched pathways in each endothelial subcluster and the associated overlap between clusters. Of note, some pathways may be classified in more than one cluster. Circle size denotes the pathway enrichment score based on the BH-adjusted p-value. A minimum score of 1.3 (i.e. FDR <0.05) was used as an inclusion threshold. Z-score color denotes the predicted activation state of the pathway based on the degree of matching between the expected and observed pattern of gene expression changes. Only pathways with z-score >+2 (activated) or <−2 (inhibited) are shown in color to highlight confident predictions. White circles denote pathways in which the activation state cannot be confidently predicted. No pathways were significantly enriched in DE genes from Cluster 6 at FDR <0.05. The identities of all significantly enriched canonical pathways are presented in *Figure 7—figure supplement 1*.

The online version of this article includes the following figure supplement(s) for figure 7:

**Figure supplement 1.** Pathway enrichment analysis of endothelial Clusters 1–7 via Ingenuity Pathway Analysis.

The fact that Cluster 1 was comprised of four distinct zones, each specific to a single timepoint, afforded a unique opportunity to assess the differential gene expression in response to EC ablation within a single cluster spanning the three distinct phases of microvascular regeneration and repair. Zones 2 and 3 (days 3 and 5) both showed high numbers of DEGs (~800–1000) compared with Zone 1 (day 0) (*Figure 8A and B*), and many of these were unique to these zones (*Figure 8C*). In contrast, there were few DEGs in Zone 4, consistent with relative normalization of gene expression profiles by 7 days post-injury. A heatmap showing the top 20 DEGs relative to Zone 1 (day 0) revealed upregulation of genes related to cell growth and response to injury in Zone 2 (*Figure 8D*), some of which were

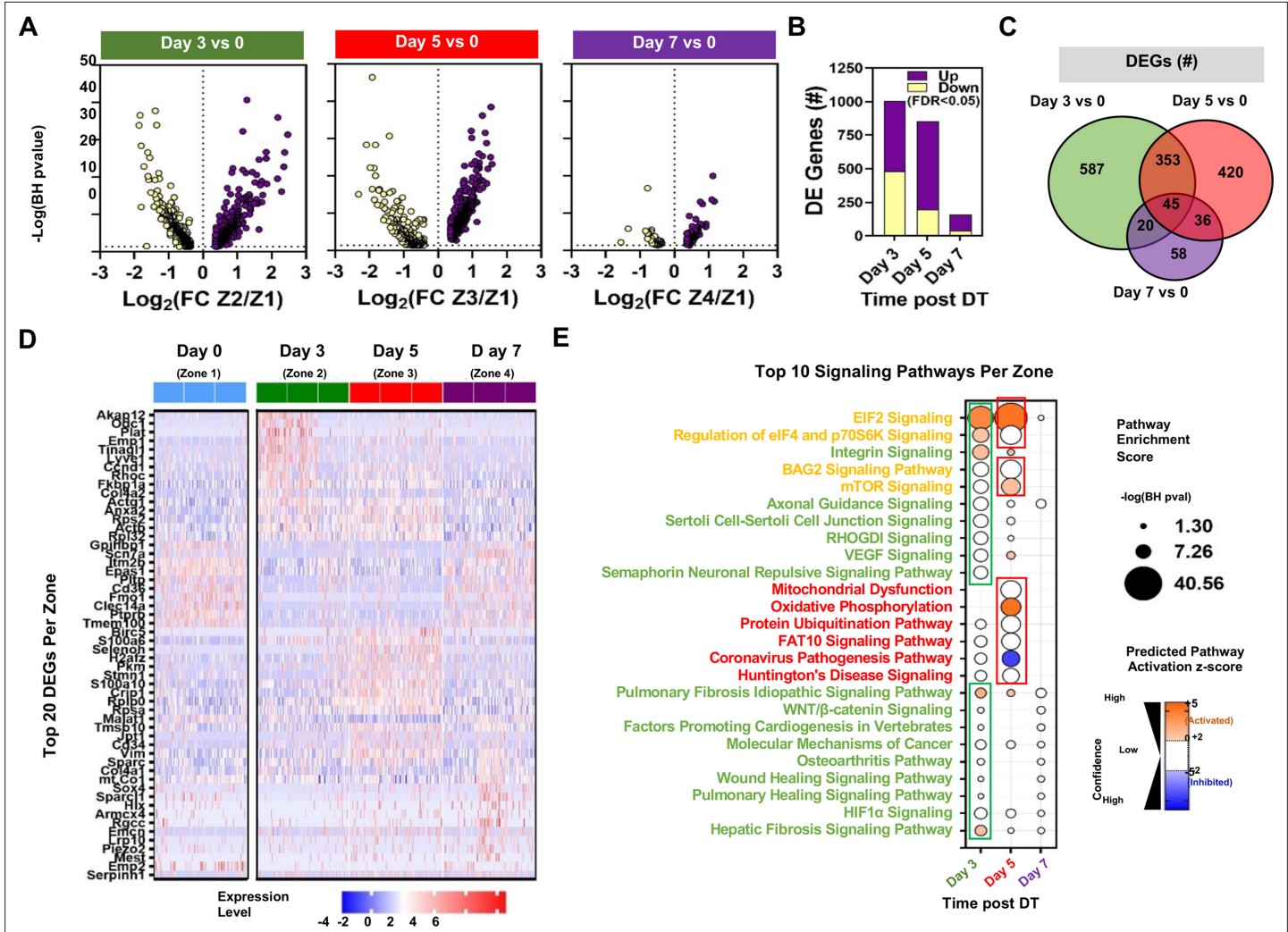

**Figure 8.** Differential expression and pathway enrichment analysis of time-dependent changes in Zones 1–4 of endothelial Cluster 1. (**A**) Volcano plots showing the distribution of differentially expressed genes according to statistical significance, magnitude, and direction of fold change (FC) at different points following administration of diphtheria toxin (DT) in mice. Only genes that passed the false discovery rate (FDR) threshold <0.05 are shown (i.e. Enrichment score –log(BH p-value)>1.3). Zones 1, 2, 3, and 4 denote day 0, day 3, day 5, and day 7, respectively. (**B**) Summary of the number of differentially expressed (DE) genes in each comparison. Purple and yellow bars are stacked to show the proportion of genes that were upregulated and downregulated, respectively. (**C**) Venn diagram shows the degree of overlap in the identity of differentially expressed genes for each comparison. (**D**) Heatmap shows expression levels of 54 genes representing the top 20 differentially expressed genes at days 3, 5, and 7 versus day 0. Expression levels (z-score of log-transformed normalized counts) are shown for a sample of 50 cells from each mouse (n=3 mice/time point). (**E**) Pathway enrichment analysis using DE gene sets from panel B with Ingenuity Pathway Analysis. The top 10 signaling pathways (by enrichment score) are shown for each time zone. Circle size denotes the pathway enrichment score based on the Benjamini-Hochberg (BH) corrected p-value. A minimum score of 1.3 (i.e. FDR <0.05) was used as an inclusion threshold. Z-score color denotes the predicted activation state of the pathway based on the degree of matching between the expected and observed pattern of gene expression changes. Only pathways with z-score >+2 (activated) or <−2 (inhibited) are shown in color to highlight confident predictions. White circles denote pathways in which the activation state cannot be confidently predicted. The identities of all significantly enriched canonical pathways in Zones 1–4 of Cluster 1 are presented in *Figure 8—figure supplement 1* with additional details in *Figure 8—figure supplement 2*.

The online version of this article includes the following figure supplement(s) for figure 8:

**Figure supplement 1.** Pathway enrichment analysis of time-dependent changes in endothelial Cluster 1 at days 3, 5, and 7 versus day 0.

**Figure supplement 2.** Chord diagram showing the activation or inhibition of genes in three canonical pathways involved in apelin signaling (PI3/Akt, mTor, Elf4/p70S6K) during the emergence of the apelin/Procr expressing general capillary (gCap) endothelial cells (ECs) at day 3.

unique to this zone (*Odc1* and *Plat*), while others were shared with Zone 3 (*Ccdn1* and *Rhoc*). Zone 3 also exhibited unique upregulation of genes involved in cell proliferation and survival (*Birc5*, *Malat1*, and *Jpt1*), as well as metabolism (*Pkm*); whereas DEGs in Zone 4 were mainly related to differentiation (*Sox4*, *Sparc1*, and *Hlx*) and cell cycle inhibition (*Rgcc*). The top 10 signaling pathways by pathway enrichment analysis showed increases EIF2, eIF4, and integrin signaling in Zone 2 ECs as well as pulmonary and hepatic fibrosis signaling (*Figure 8E*). There was further enrichment of EIF2 and mTOR signaling in Zone 3 (day 5) along with unique upregulation in oxidative phosphorylation. In contrast, there was no significant predicted activation of signaling pathways in Zone 4 (day 7) compared with the baseline at day 0 (Zone 1). The identities of all significantly enriched canonical pathways are presented in *Figure 8—figure supplement 1*. Additionally, in *Figure 8—figure supplement 2* we show changes in the expression of genes related to the apelin signalling pathways in Zone 2 of Cluster 1 (i.e. day 3 versus day 0) as a chord diagram. During the emergence of apelin expressing gCap ECs, the three major pathways mediating apelin signaling, PI3/Akt, mTor, Elf4/p70S6K, were all strongly activated.

We next explored cell-cell interactions using NicheNet, focusing on endothelial receiver populations during the critical period of endothelial regeneration. At day 3, only the transitional ECs of Cluster 1 (Zone 2) met the criteria for an endothelial 'receiver' population (i.e. >50 DEGs versus day 0) (*Figure 9A and B*) largely representing *Apln/Procr* co-expressing gCap ECs. These cells received signals from 21 clusters representing all major cell populations (stromal, myeloid, lymphoid, epithelial, and endothelial). The top four predicted ligands based on Pearson correlation coefficients were Occludin (*Ocln*), FAT Atypical Cadherin 1 (*Fat1*), Plexin B2 (*Plxnb2*), and Ephrin B1 (*Efnb1*). These all mediate signaling by direct cell-cell contact and play major roles in developmental biology, including regulation of proliferation, migration, and differentiation. *Ocln* and *Efnb1* were associated with interactions between EC subtypes, whereas *Plxnb2* mediates cell-cell signaling between ECs and non-EC clusters including pericytes, monocyte/macrophages, and epithelial cells. In contrast, *Fat1* was restricted to interactions between EC and stromal cells, specifically pericytes and smooth muscle cells. On day 5, there was a substantially greater diversity in cell-cell interactions, with five endothelial receiver populations responding to ligands from 31 cell clusters representing all cell populations (*Figure 9C and D*). Again, transitional gCap ECs (Cluster 1, Zone 3) exhibited the strongest predictions for the top ligands which, with the exception of *Fat1*, remained the same as foday 3. The other endothelial receiver clusters showed considerable differences in their top ligands, likely reflecting the distinct pathways involved in the differentiation of Cluster 1 ECs into these specialized subpopulations during the microvascular repair.

## ALI resolution and survival is dependent on apelin signaling

An analysis of a publicly available atlas of the aging lung (*Angelidis et al., 2019*) revealed a significant reduction in lung apelin expression in mice (*Figure 10A*) which was confirmed in the Cdh5-DTR double transgenic mice by RT-qPCR (*Figure 10B*). Interestingly, aging reduced survival post-DT-induced injury with only 24% of 52-week-old Cdh5-DTR mice surviving to day 7 post-DT compared to 82% survival for 12-week-old mice (p<0.01) (*Figure 10C*). Moreover, when young Cdh5-DTR mice were treated with an apelin receptor antagonist, ML221, all mice succumbed to DT-induced acute lung injury by 5 days (*Figure 10C*), consistent with a failure of lung microvascular repair.

## Discussion

We have established that selective ablation of lung ECs results in severe ALI with many features of ARDS seen in various clinic contexts, including COVID-19. Remarkably, the loss of more than 70% of lung ECs was compatible with survival by virtue of an efficient endogenous regenerative response that resulted in near complete restoration of the lung microvascular structure and function in just one week. This model underscores the importance of the endothelium in the pathogenesis of ALI and affords a unique opportunity to explore the mechanisms underlying EC regeneration and microvascular repair, which is critical for ALI resolution. Using single-cell transcriptomic analysis, we have identified novel regenerative EC populations emerging in a tightly scripted temporal sequence after EC injury. For the first time, we show robust apelin expression within gCap, stem-like ECs that give rise to apelin receptor-positive, highly proliferative progenitors which are responsible for the replenishment

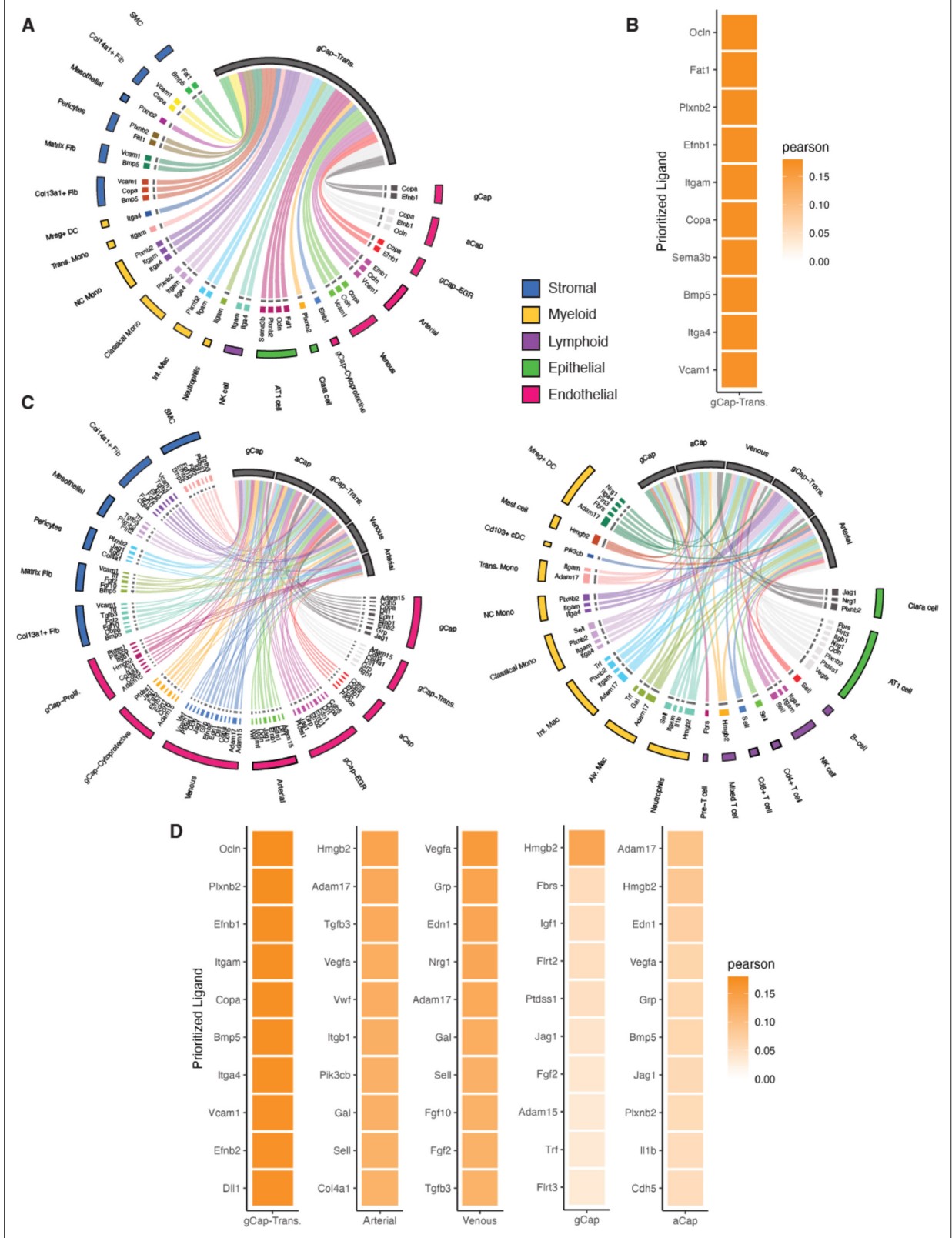

**Figure 9.** Cell Signaling inferences using NicheNet. Circos plots showing color-coded sender populations (outer circle) indicating the 10 top genes signaling for each receiver population (shown in gray). (**A**) At day 3, only one population, general capillary (gCap)-transitional Cluster 1, met the criteria for a 'receiver' and this cluster received signals form 21 separate cell clusters representing five distinct cell populations (stromal, myeloid, lymphoid, epithelial, and endothelial). (**B**) Heatmap shows the Pearson's correlation coefficients of the top 10 ligand-receptor interactions for the gCap endothelial

*Figure 9 continued on next page*

*Figure 9 continued*

cell (EC) receiver population. (**C**) At day 5, there was a substantially greater diversity in cell-cell interactions, with five endothelial receiver populations responding to signals from 31 cell clusters. (**D**) Heatmaps of Pearson's correlation coefficients of the top 10 ligand-receptor interactions for each receiver population.

of all depleted EC pools, including the highly specialized aerocytes (aCap ECs) that reform the air-blood barrier.

In the normal uninjured lung, we confirmed that the expression of apelin identified an EC population corresponding to alveolar aCap ECs, or 'aerocytes,' as previously described by *Gillich et al., 2020*. These were characterized as large, highly specialized cells forming the air-blood barrier, thereby playing an important structural role in gas exchange. This population also appears to be analogous to an EC population described earlier in close apposition to alveolar type 1 epithelial cell and expressing high levels of *Car4* (*Niethamer et al., 2020*; *Vila Ellis et al., 2020*), one of five genes typically expressed by aCap ECs (*Gillich et al., 2020*; *Vila Ellis et al., 2020*). We also describe, for the first time, the emergence of novel EC populations after EC injury which, based on the expression of typical marker genes, originate from gCap ECs. In particular, we have identified a gCap EC population that, remarkably, exhibits de novo expression of apelin together with gCap markers such as CD93 (with the exception of the apelin receptor) demonstrating for the first time apelin expression in gCap ECs. This population was also characterized by the unique expression of two other genes, *Procr* (protein C receptor) and *Angpt2* (angiopoietin 2), that have both been implicated in angiogenesis and vascular repair. Angiopoietin 2 is an endogenous antagonist of the Tie2 receptor that is strongly expressed in endothelial tip cells (*Felcht et al., 2012*) and is instrumental in the initiation of angiogenic response (*Maisonpierre et al., 1997*), whereas *Procr* has recently been reported to be a marker of bipotent resident vascular endothelial stem cells that have the capacity to regenerate ECs as well as pericytes (*Yu et al., 2016*), the two cell types required for the genesis of stable neovessels. Interestingly, the expression of *Procr* also identifies a subpopulation of CD34[+] hematopoietic stem cells (HSCs) with markedly greater proliferative and bone marrow engraftment potential (*Balazs et al., 2006*). Together, this is consistent with the stem cell nature of this transient population of endothelial stem cells which initiates the regenerative response after EC ablation.

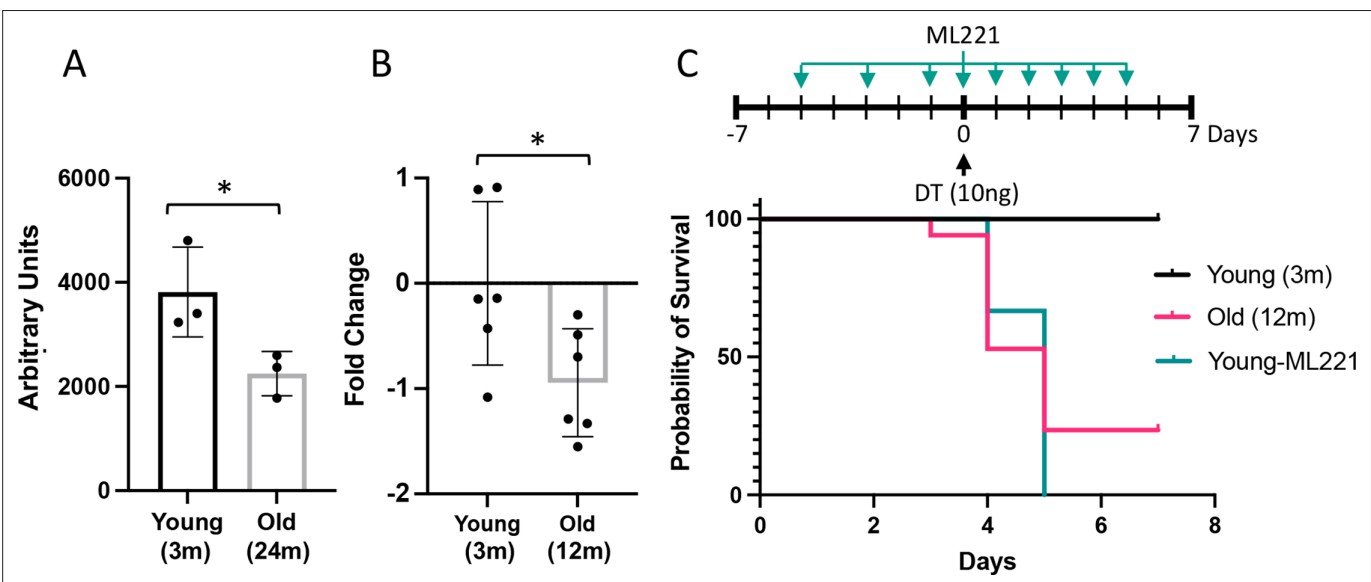

**Figure 10.** Impairment or inhibition of the apelin pathway leads to failure of recovery after endothelial cell (EC) ablation. (**A**) Expression of apelin from a publicly available dataset (31) in young (3 months) versus old (24 months) mice (*=p<0.05, unpaired *t*-test). (**B**) Apelin expression assessed by qRT-PCR expressed as fold-change relative to mean value in young (3 months) Cdh5-DTR binary transgenic mice (*=p<0.05, unpaired *t*-test). (**C**) Survival after intra-tracheal (IT) instillation of diphtheria toxin (DT) of either 3 (Young) or 12-month-old (Old) Cdh5-DTR with or without administration of the apelin antagonist, ML221 (10 mg/kg IP). Survival of old mice or mice receiving ML221 was significantly different from young control mice (p<0.05, Mantel-Cox log-rank test). n=11 Young-DT, n=17 Old-DT, n=7 Young-DT+ML221, n=3 Control ML221. Data presented as mean ± SD.

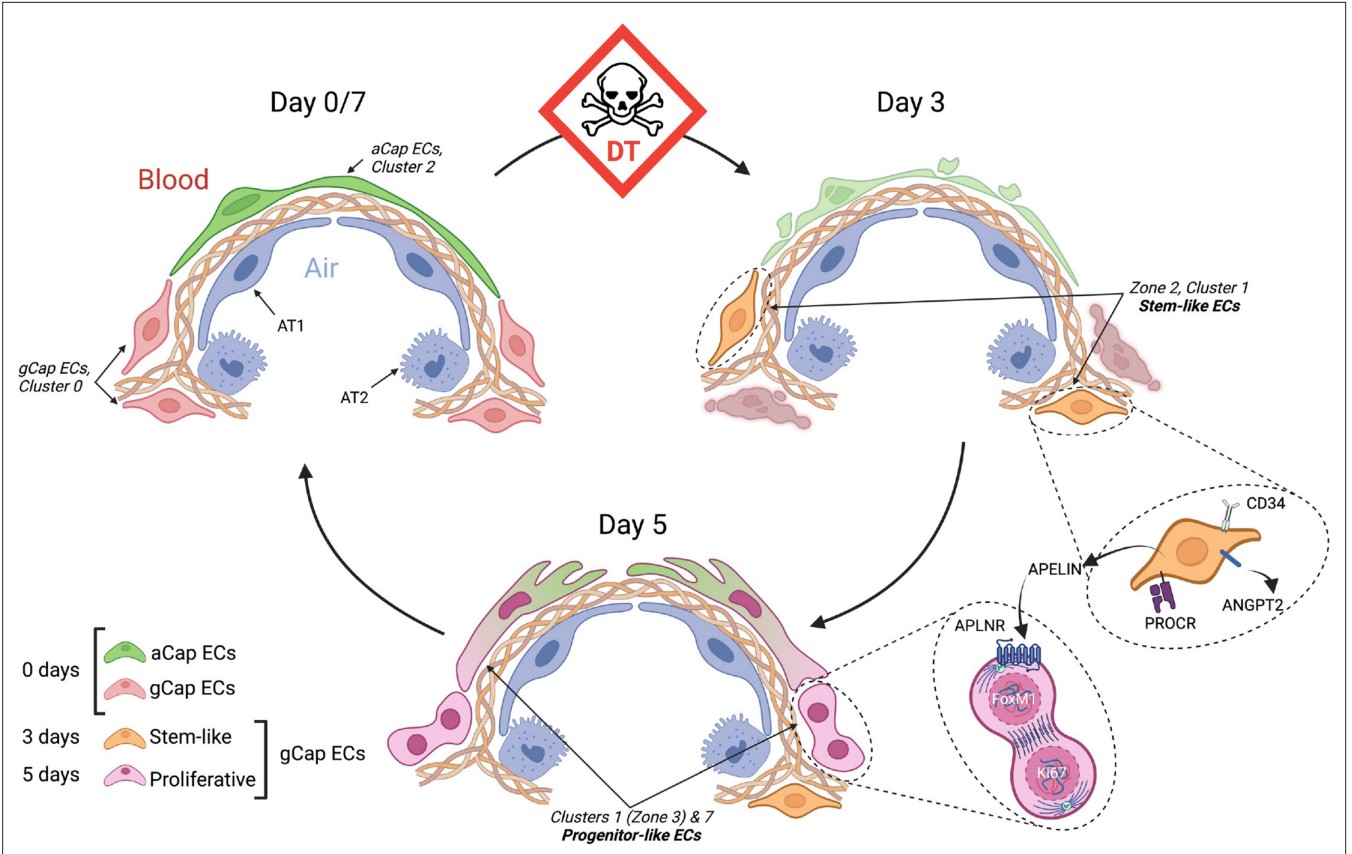

**Figure 11.** Schematic of endothelial cell (EC) populations contributing to microvascular repair. At baseline (day 0), there are two main alveolar groups of capillary ECs: larger apelin-positive aerocytes (aCap) ECs, termed aerocytes, that play a key structural role in forming the air-blood barrier; and smaller apelin receptor-expressing general capillary (gCap) ECs, which are found in the thicker regions at the corners of the alveoli. After diphtheria toxin (DT)-induced EC ablation, there is a marked depletion of both EC populations and the appearance of novel transitional and transient populations. At day 3, there is the appearance of stem-like gCap ECs that paradoxically express apelin, but not its receptor, and are characterized by various stem and progenitor cell markers but show no evidence of proliferation. By day 5, these transition to ECs expressing *Aplnr* which have a strong proliferative phenotype, as evidenced by Forkhead box M1 (*Foxm1*) and *Mki67* expression, and then rapidly replenish depleted EC pools, including aCap ECs, by day 7. This transition is orchestrated by the interaction of apelin with its receptor as a critical mechanism in lung microvascular regeneration after EC injury. AT1, alveolar type –1 epithelial cell; AT2, alveolar type-2 epithelial cell; APLNR, apelin receptor; ANGPT2, angipoietin 2; PROCR, protein C receptor.

Created with BioRender.com.

Apelin-positive, gCap ECs showed no evidence of proliferation at 3 days post-injury; however, by 5 days they transitioned to a highly proliferative phenotype with expression of apelin being replaced by its receptor, consistent with a dynamic role for the apelinergic system in driving a transition from the endothelial tip- to stalk-like, proliferative progenitor cells (*Palm et al., 2016*). At the same time, a very similar stalk-like progenitor population emerged within the transient EC Cluster 7. Both these populations expressed *Foxm1*, which has previously been implicated in lung microvascular repair (*Zhao et al., 2014*), and exhibited similar proliferative gene expression signatures; however, Cluster 7 showed a greater density of *Foxm1* and *Mki67* expressing proliferative ECs. Finally, the pivotal role of apelin in EC regeneration was confirmed by the failure of resolution of ALI after DT-induced lung injury and excessive mortality in older mice exhibiting reduced expression of apelin and with the administration of a selective receptor antagonist. The proposed mechanism of microvascular endothelial repair is illustrated in *Figure 11*.

There are some limitations of the present study which include the lack of direct spatial mapping of EC populations identified by single-cell transcriptomics analysis; however, this was mitigated in part by immunofluorescence staining taking advantage of the unique co-expression of apelin and PROCR by the stem-like, gCap ECs at 3 days post EC ablation. As well we were not able to perform lineage

tracing studies to define the origin of the endothelial stem- and progenitor-like cells that were responsible for repopulating all depleted EC populations given the difficulties of doing this in a double transgenic model compounded by the very transient nature of the populations of interest. However, consistent with our findings it was previously demonstrated by lineage tracing that only gCap ECs exhibited evidence of proliferation after lung injury, which was interpreted as evidence that regenerative stem cells would reside only within this population (**Gillich et al., 2020**). Moreover, the use of single-cell transcriptomics allowed us to reliably identify aCap and gCap ECs at each timepoint during endothelial repair using a panel of marker genes that distinguish between these distinct populations (**Gillich et al., 2020**; **Schupp et al., 2021**).

Speculation and Ideas: *Procr* has been suggested to be a marker of primitive, endothelial-like hematopoietic precursors (i.e. pre-HSCs) (**Lan, 2017**) that give rise to both ECs and HSCs during development (**Dzierzak and Bigas, 2018**), playing a critical role in the early genesis of both blood and blood vessels. *Procr* has also been reported to be expressed by ~90% of bone marrow side population cells (**Balazs et al., 2006**), which are thought to have important regenerative activity. However, *Procr* was not expressed by any EC population in the healthy lung, and only appeared transiently in the gCap endothelial stem- and progenitor-like cells during the microvascular repair. This suggests that lung vascular injury results in a re-activation of a fundamental vascular stem cell that is central to both vascular development and repair. Whether this also occurs in other vascular beds after the endothelial injury is unknown; however, the protean nature of the expression of this stem-cell marker in different cell types would support its role as a general mechanism in vascular regeneration and repair. The unique co-expression of *Procr* and apelin in this gCap EC population also allows for the identification and isolation of these regenerative cells to explore the mechanisms underlying their generation post-EC injury. Indeed, on day 3 post-DT-induced ablation, ~15% of all lung ECs expressed *Procr* making their isolation feasible using commercially available kits for PROCR. As well, the therapeutic potential of these cells could be explored using adoptive cell therapy, for example, in syngeneic preclinical models of ALI. However, for translation to clinical use, one would need to develop a cell product suitable for human use that could be scaled. To this end, small molecule libraries could be screened for the ability to induce expression of apelin and EPCR, with the intent of generating endothelial stem-like cells in cultured ECs. As a readily available source of autologous ECs, colony-forming ECs derived from circulating monocytes (**Yoder et al., 2007**) would be well suited for this purpose using *Apln* and *Procr* reporter genes and high throughput technology.

Thus, we have delineated the cellular and molecular processes involved in rapid and complete microvascular regeneration using a new model of ALI induced by targeted EC ablation. Single-cell transcriptomic analysis has revealed a novel gCap endothelial stem-like population expressing apelin and *Procr* and emerging soon after endothelial injury, subsequently giving rise to *Foxm1*-positive progenitor-like ECs expressing the apelin receptor. These highly proliferative ECs were responsible for replenishing all depleted EC populations leading to the rapid resolution of the ALI phenotype by an apelin-dependent mechanism. These findings highlight the critical role of apelin signaling in regulating the activity of regenerative cells that mediate lung microvascular repair and provide insights for the development of novel regenerative strategies for the treatment of ALI and ARDS.

## Methods
### Transgenic models

All animal procedures were approved by the University of Ottawa Animal Care Ethics Committee in agreement with guidelines from the Canadian Council for the Care of Laboratory Animals under protocol OHRI-2747. Transgenic animals were obtained from Jackson's Laboratories including mice with Cre-inducible DTR flanked by loxP sites (C57BL/6-*Gt(ROSA)26Sor^{tm1(HBEGF)Awai}*/J) and expressing Cre-recombinase under the EC restricted promoter, VE-cadherin (Cdh5) (B6.FVB-Tg(Cdh5-cre)7Mlia/J) (Stock numbers: 007900 and 006137, respectively). Stocks were maintained by crossing homozygous animals. Only male animals 10–12 weeks of age were used for experiments in this manuscript, unless otherwise specified. All animals were genotyped using primer sequences provided by Jackson's Laboratory.

## DT administration

Stocks of DT from *Corynebacterium diphtheriae* (Sigma-Aldrich, Oakville, ON, Canada) were prepared (1 mg/mL) in sterile distilled water and stored in single-use aliquots at –20 °C. Animals were anesthetized with ketamine (100 mg/Kg i.p.) and xylazine (10 mg/Kg i.p.) and DT was delivered by intratracheal (IT) instillation (10 ng in 50 µLs, unless otherwise specified) with a 1 mL syringe and measured with a micropipette. Prior to toxin administration, a water droplet was used to indicate the proper positioning of the catheter into the orotracheal cavity and was deemed successful by the movement of the droplet with the breathing of animals. Control animals received 50 µLs of 0.9% sterile saline.

## Lung dissection and single-cell dissociation

Animals were anesthetized with ketamine (100 mg/kg) and xylazine (10 mg/kg) (i.p.) and given 10 mU/g heparin sodium (i.p.) (LEO Pharma Inc, Thornhill, ON, Canada). The inferior vena cava was exposed surgically, and the animals were bled and then 25 U/mL heparin in 10 mLs of sterile 0.9% saline was flushed through the lungs via the pulmonary artery until they were cleared of all evidence of red blood cells. Lungs were rinsed in PBS, cut into smaller pieces, and placed into gentleMACS C tubes (Miltenyi Biotech, Bergisch Gladbach, Germany) containing 2.5 mLs DPBS to which 2.5 mLs of digestion enzyme mix were added. This mix included 2500 U Collagenase I, 30 U Neutral Protease/Dispase (Wothington Biochem., Lakewood, NJ, USA), and 500 U Deoxyribonuclease (Sigma-Aldrich, Oakville, ON, Canada) in 1 x Dulbecco's PBS (Thermo Fischer Scientific, Burlington, ON, Canada), and was made fresh for each experiment and kept on ice. GentleMACS C tubes were placed into a temperature-regulated gentleMACS Octo Dissociator (Miltenyi Biotech, Bergisch Gladbach, Germany) and underwent mechanical dissociation according to a custom mouse lung program at 37 °C for 30 min. Dissociated tissue was passed through a pre-wetted 75 µm filter (Thermo Fischer Scientific, Burlington, ON, Canada), re-suspended with an additional 5 mLs PBS, and treated with 200 µLs 0.5 M EDTA (Thermo Fischer Scientific, Burlington, ON, Canada). Pelleted cells were re-suspended in RBC-lysis buffer (Thermo Fischer Scientific, Burlington, ON, Canada) for 3 min at room temperature. Final cell pellet was re-suspended in 5 mLs PBS and cell counts and viability was performed using a Countess automated counter (Thermo Fischer Scientific, Burlington, ON, Canada).

## Flow cytometric analysis

Single-cell suspension was added to a v-bottom 96-well plate (Corning, NY, USA) $0.5–1 \times 10^6$ cells/well. Live/Dead fixable staining assay was performed as instructed by product information (Thermo Fischer Scientific, Burlington, ON, Canada, L34973). Cells were blocked in FcR blocking reagent (Miltenyi Biotech, Bergisch Gladbach, Germany, 130-092-575) for 15 min and incubated in 1:100 dilution PE-CD-144 (BD Biosciences, Mississauga, ON, Canada, 562243), PE-Dazzle-CD-34 (BioLegend, San Diego, CA, 128615) and APC-Fire-CD-31 (BioLegend, San Diego, CA, USA, 102433) for 30 min. Cells were washed in FACS buffer (PBS, 1% BSA, 1 mM EDTA) and fixed in 2% PFA for 10 min. 500 µLs of cells in FACS buffer were passed through a 40 µm mesh into flow tubes (BD Biosciences, Mississauga, ON, Canada). Samples were analyzed on a BD LSR Fortessa using BD FACSDIVA software for compensation and gating (Beckton Dickinson Biosciences, Franklin Lakes, NJ, USA). Further cytometric analyses were conducted on FlowJo v.10.6.2 (FlowJo LLC, Ashland, OR, USA).

## Immunofluorescence staining

To compare the level of apoptotic cell death, active caspase-3 was stained following the steps below. After removal of paraffin, PBS washing, and microwave treatment in citrate buffer (pH 6.0; H3300-250, Vector laboratories), neutral buffered formalin-fixed sections were blocked with 5% BSA (800–095-EG, Wisent) in wash buffer containing PBS with 0.25% TritonX-100 (T8787, Sigma-Aldrich). Diluted to 4 µg/mL using 1% BSA (in wash buffer), human/mouse active caspase-3 polyclonal antibody (AF835; R&D Systems) were incubated overnight at 4 °C in a humidified chamber. Binding of the caspase-3 antibody was detected using Goat anti-rabbit secondary conjugated with Alexa Fluor 594 (A11012, Invitrogen). Counterstain was performed using DAPI (4,6-diamidino-2-phenylindole) at 5 µg/mL for 10 min, before applying Vectorshield PLUS mounting medium (H1900, Vector Laboratories). Sections were imaged using an upright epifluorescence microscope (imager M2, Zeiss). Fluorescent images were processed with FIJI open-source software (https://github.com/fiji; *Schindelin et al., 2012*).

## Double IF staining for DTR and CD144

Paraffin sections (5 µm thickness) were deparaffinized and rehydrated in a sequence of xylene, ethanol, and ddH$_2$O washes. Heat-induced antigen retrieval with Antigen Retrieval Reagent-Basic (CST013, R&D Systems) was performed before staining with DTR and CD144 sequentially. First, sections were blocked 5% BSA in PBS with 0.25% Tween 20 (PBST) and incubated with 1:100 goat anti-human HB-EGF (AF-259-SP, R&D Systems, Oakville, ON, Canada) over-night at 4 °C. Following washes in PBS (with 0.25% Triton-X-100), sections were incubated with Donkey anti-goat Alexa 647 secondary (Thermo Fischer Scientific, Burlington, ON, Canada) for 1 hr at room temperature. For the biotinylated primary antibody against CD144, tissue endogenous biotin was blocked. Streptavidin/biotin blocking kit (Vector Laboratories; SP2002) was used with minor modifications to enhance the specificity of Biotinylated anti-CD144 staining. Briefly, streptavidin was diluted using PBST (PBS with 0.1% Tween-20). Slides were incubated at room temp for 15 min, followed by washing with PBST for three times (each for 10 min). Biotin diluted with PBST was applied to samples and incubated at room temperature for 45 min. Slides were washed 3 times with PBST (each for 10 min). After blocking of tissue-derived biotin, Biotinylated CD144 antibodies (BAF1002, R&D systems, ON, Canada) were incubated with the tissue sections for over-night at 4 °C. Binding of the Biotinylated anti-CD144 was detected using streptavidin conjugated with Alexa Fluor 568 (S11226, Invitrogen; 1:300 dilution of 2 mg/mL stock). Counterstain was performed using DAPI at 5 µg/mL for 10 min, before mounting with Vectorshield PLUS mounting medium (H1900, Vector Laboratories). Sections were imaged using an inverted confocal microscope (LSM900 with an Axio Observer Z1/7 stand, Zeiss). using a 40 x objective (C-Apochromat, NA1.2, water immersion). Two separate tracks were arranged in Zen blue software to image DTR (Alexa647) and CD144 (Alexa568) with GaAsP-PMT detectors using 6.5% 640 nm laser and 561 nm laser, respectively. Pinholes were kept the same at 42 µm. Acquired images were processed with FIJI open-source software (https://github.com/fiji; *Schindelin et al., 2012*).

## Evans blue assay

A solution of 0.5% Evans blue (Eb) (Sigma-Aldrich, Oakville, ON, Canada) was prepared fresh in 0.9% sterile saline and passed through a 22 µm filter. 100 µL of Eb solution was injected via the tail-vein and animals were allowed to rest for 2 hr before being sacrificed and bled. A catheter was inserted into the pulmonary artery and flushed with 10 mL of saline at a constant pressure (~25 mmHg). Organs were collected and allowed to air dry for 2 hr. Dried tissue was weighted and placed in centrifuge tubes with 500 µL of formamide (Sigma-Aldrich, Oakville, Ontario, Canada), incubated at 55 °C for 48 hr, and then centrifuged at 12,000 g for 20 min. Absorbance was measured in a plate reader at 610 nm wavelength. Concentration of Eb was cross-referenced to the standard curve and concentration per tissue weight was analyzed for each tissue and condition collected.

## Lung Injury scoring

Lung samples were collected to produce formalin fixed paraffin embedded blocks, and tissue sections were cut to stain with hematoxylin and eosin (H&E), as previously described (*Chaudhary et al., 2018*). Lung injury scoring was performed following the American Thoracic Society's guidelines by evaluating the following parameters: neutrophils in the alveolar space, neutrophils in the interstitial space, hyaline membranes, proteinaceous debris filling the airspaces, and alveolar septal thickening (*Matute-Bello et al., 2011*). Lung injury scoring was performed by a blinded reviewer.

## MULTI-seq cell barcoding

Barcoding of individual biological samples was performed as described by *McGinnis et al., 2019b*. In summary, dissociated lung cells (0.5 × 10$^6$ cells per sample) were suspended in 150 µLs solution containing a 1:1 molar ratio (200 nM) of anchor and barcode oligonucleotide containing a unique sequence for each of the 12 samples to be processed. Samples were incubated for 13 min at room temperature with gentle mixing every 3–5 min. Next, a co-anchor (200 nM) was added to stabilize barcodes within the membrane and incubated for additional 5 min. Cells were washed twice in PBS and cell counts were measured using a Countess automated counter (Thermo Fischer Scientific, Burlington, ON, Canada) and viability was measured based on the ratio of cells staining with trypan blue (Thermo Fischer Scientific, Burlington, ON, Canada). Equal ratio of cells from each 12 barcodes was

pooled and 1000 cells/µL were further processed through the 10x-Genomics pipeline. Only samples with viability >85% were used.

## Processing single-cell RNA sequencing libraries

RNA library construction with 10 x Genomics Single-cell 3' RNA sequencing kit v3 was processed as previously described (*Cook and Vanderhyden, 2020*). Gene expression libraries were prepared as per the manufacturer's recommendations. Libraries for 48,347 cells were sequenced using a NextSeq500 (Illumina) with a mean of 8550 reads per cell, and a median of 1560 UMIs and 845 genes per cell. CellRanger v4.0 software (10 x Genomics) was used to process raw sequencing reads with the mm10 reference transcriptome and with additional manual annotation of the DTR transgene. MULTI-seq barcode libraries were further trimmed to 28 bp using Trimmomatic v.0.39 (https://github.com/timfl-utre/trimmomatic; *Bolger et al., 2014*).

## Demultiplexing, doublet removal, and quality control

Barcodes were demultiplexed using the R package deMULTIplex (*McGinnis et al., 2019b*) (https://github.com/chris-mcginnis-ucsf/MULTI-seq). Cells lacking barcodes or with multiple barcodes (doublets) were excluded from further analysis. Cells underwent an additional doublet removal step using the R package DoubletFinder (*McGinnis et al., 2019a*) (https://github.com/chris-mcginnis-ucsf/DoubletFinder) and scDblFinder (https://github.com/plger/scDblFinder). Quality control was performed using the R package Seurat (*Butler et al., 2018*) v.3.1.5 (https://github.com/satijalab/seurat). Cells with a high proportion (>30%) of mitochondrial transcripts and those with low complexity (<200 detected genes) were excluded from the final matrix (*Figure 3—figure supplement 1*). A total of 21,665 cells were used for downstream analyses. Data were log normalized and variable genes were detected using 'vst' method. Integrated analysis on top 3000 genes was conducted under 'SCT' method where cell cycle and mitochondrial content and treatment condition were regressed out prior to the calculation of PCA and UMAP on the first 40 principle components. Cell clusters were characterized using an automated annotation tool (*Tan and Cahan, 2019*) and by cross-referencing differential gene expression of individual clusters to previously characterized lung cells of the *Tabula Muris* cell atlas (*Tabula Muris Consortium et al., 2018*). The identity of the various lung cell clusters was further confirmed by the assessment of the expression of cell-specific genes. Subset analysis was performed on the following bulk populations: endothelial, stromal, lymphoid, myeloid, and epithelial where cell numbers allowed. Subset analysis followed the same integration pipeline during original processing, and high-resolution cell identities were recombined to produce a global UMAP with high-resolution nomenclature (*Figure 3A*).

## Differential expression analysis

Differential gene expression analysis was conducted using R package Muscat (*Crowell et al., 2020*) multi-sample multi-group scRNA-seq analysis tools (*Crowell et al., 2023*). Standard workflow was performed by generating pseudobulk expression profiles for each cluster and testing for differential expression between experimental groups/conditions using default parameters unless otherwise specified.

## Automated cell classification

The different cell populations identified in our study were cross-referenced with pre-annotated lung cells from the (*Tabula Muris Consortium et al., 2018*) using the R package singleCellNet (https://github.com/pcahan1/singleCellNet; *Tan and Cahan, 2019*).

## Cell type prioritization

To determine cells most affected during our different conditions in relation to control samples, we have employed a machine learning model to predict cells that become more separable during treatment based on their molecular measurements. For this, we used the R package Augur (*Skinnider et al., 2021*) (https://github.com/neurorestore/Augur).

## RNA velocity and trajectory inference

To evaluate RNA velocity and perform trajectory inference we used CellRank (*Lange et al., 2022*). Briefly, CellRank (version 1.5.2.dev139+g44b213e) was used as a python package (python version

3.8.13). CellRank requires results from scVelo as input to infer macrostates typically include initial and terminal states. To fulfill this need, a dynamic model from scVelo (version 0.2.4) (*Bergen et al., 2020*) was used to estimate RNA velocity.

## Transcription factor analysis

Transcription factor activity was predicted using the R package decoupleR (version 2.0.1) (*Badia-I-Mompel et al., 2022*). Using the weighted mean method in decoupleR, we compared our single-cell gene expression data and inferred TF and their targets based on the curated network DoRothEA (*Garcia-Alonso et al., 2019*). The top 25 TFs with the most variable activity between endothelial populations were depicted with their average activity by cluster in *Figure 6—figure supplement 2*.

## Ingenuity pathway analysis

Pathway enrichment analysis was performed in Ingenuity Pathway Analysis (IPA) web-based software application using standard workflow in the core analysis function to identify the canonical pathways from the IPA library that were most significantly enriched in the differentially expressed gene sets under investigation (*Krämer et al., 2014*).

## Cell signaling inference

To evaluate potential cell-cell signaling cues within the lung during ALI resolution we used the R software package NicheNet (*Browaeys et al., 2020*). NicheNet takes prior knowledge of cell signaling interactions from data bases of ligand-receptor, cell signaling, and gene regulatory networks. We sought to evaluate cell signaling at day 3 and day 5 of ALI resolution compared to control (day 0) conditions. To identify potential receiver clusters, we set a threshold of >50 DEGs at each timepoint compared to control (day 0) conditions, identifying cells that had been strongly affected by DT injury. We chose to retain all cell types as potential sender cells of the inferred prioritized ligands. Following the pipeline established by NicheNet we identified the top predicted ligands affecting each receiver cluster based on the Pearson correlation coefficient, evaluated by NicheNet. The circos plots (*Figure 9*) identify which cells are predicted to send these top ligands during ALI.

## Apelin inhibition

The role of the apelin pathway in lung microvascular repair was assessed using a selective apelin receptor antagonist, ML221 (Tocris, 10 mg/kg) (*Maloney et al., 2012*) dissolved in DMSO delivered by intraperitoneal injection as previously described (*Ishimaru et al., 2017*).

## Code availability

Code used to generate scRNAseq analysis is available at https://github.com/rsgodoy/Single-Cell-Transcriptomic-Atlas-of-Lung-Microvascular-Regeneration-After-Targeted-EC-Ablation (copy archived at *Godoy, 2023*).

## Statistical analysis

Statistical analysis was performed using Prism v.8.4.2 (GraphPad Software). Results are expressed as means ± SEM. For scRNA-seq experiments, three animals were used per group at each time point and statistical analyses were performed according to the recommendations specified for their analytic packages used as described in the Results section and figure legends. The specific statistical analysis used in each experiment is presented in the corresponding figure legends.

# Acknowledgements

We would like to thank Dr. Saad Khan, Dr. Maria Hurskainen, and Dr. Ivana Mizikova for procedural guidance and advice and Dr. Bernard Thébaud for his advice on the manuscript. We would also like to thank the Ottawa Hospital Research Institute's core facilities StemCore Laboratories and the Ottawa Bioinformatics Core Facility, and the University of Ottawa's Cell Biology and Image Acquisition core facility. This work was supported by a Foundation award from the Canadian Institutes of Health Research (FDN – 143291) to DJS. NDC acknowledges scholarship support from the Canadian Institutes of Health Research.

## Additional information

### Funding

| Funder | Grant reference number | Author |
|---|---|---|
| Canadian Institutes of Health Research | FDN - 143291 | Duncan J Stewart |

The funders had no role in study design, data collection and interpretation, or the decision to submit the work for publication.

### Author contributions

Rafael Soares Godoy, Conceptualization, Data curation, Software, Formal analysis, Validation, Investigation, Visualization, Methodology, Writing – original draft, Writing – review and editing; Nicholas D Cober, Data curation, Software, Formal analysis, Investigation, Visualization, Methodology, Writing – review and editing; David P Cook, Data curation, Software, Formal analysis, Methodology; Emma McCourt, Data curation, Software, Validation, Visualization; Yupu Deng, Katelynn Rowe, Investigation, Methodology; Liyuan Wang, Data curation, Software, Validation, Investigation, Visualization, Methodology; Kenny Schlosser, Data curation, Software, Formal analysis; Duncan J Stewart, Conceptualization, Resources, Data curation, Supervision, Funding acquisition, Methodology, Writing – original draft, Project administration, Writing – review and editing

### Author ORCIDs

Nicholas D Cober ⓘ http://orcid.org/0000-0001-8061-806X
David P Cook ⓘ http://orcid.org/0000-0001-7639-6724
Emma McCourt ⓘ http://orcid.org/0000-0003-2796-9279
Duncan J Stewart ⓘ http://orcid.org/0000-0002-9113-8691

### Ethics

All animal procedures were approved by the University of Ottawa Animal Care Ethics Committee in agreement with guidelines from the Canadian Council for the Care of Laboratory Animals under protocol OHRI-2747.

### Decision letter and Author response

Decision letter https://doi.org/10.7554/eLife.80900.sa1
Author response https://doi.org/10.7554/eLife.80900.sa2

## Additional files

### Supplementary files
• MDAR checklist

### Data availability

Sequencing data have been deposited in GEO under accession codes GSE211335.

The following dataset was generated:

| Author(s) | Year | Dataset title | Dataset URL | Database and Identifier |
|---|---|---|---|---|
| Godoy RS, Cook DP, Cober ND, McCourt E, Deng Y, Wang L, Schlosser K, Rowe K, Stewart DJ | 2022 | Single Cell Transcriptomic Atlas of Lung Microvascular Regeneration after Targeted Endothelial Cell Ablation | https://www.ncbi.nlm.nih.gov/geo/query/acc.cgi?acc=GSE211335 | NCBI Gene Expression Omnibus, GSE211335 |

The following previously published datasets were used:

| Author(s) | Year | Dataset title | Dataset URL | Database and Identifier |
|---|---|---|---|---|
| Simon LM, Schiller HB | 2019 | Multi-modal analysis of the aging mouse lung at cellular resolution | https://www.ncbi.nlm.nih.gov/geo/query/acc.cgi?acc=GSE124872 | NCBI Gene Expression Omnibus, GSE124872 |
| Zhang L, Gao S, White Z, Dai Y, Malik AB, Rehman J | 2020 | Single-Cell Transcriptomic Profiling of subpopulations of vascular endothelial cells following acute lung injury | https://www.ncbi.nlm.nih.gov/geo/query/acc.cgi?acc=GSE148499 | NCBI Gene Expression Omnibus, GSE148499 |

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
