## [Editor Report]

The manuscript by Godoy and colleagues is an important contribution to the understanding of how lung endothelial regeneration progresses following endothelial ablation. The novelty and elegance of this study are rooted in the regional and specific ablation of lung endothelial cells using diphtheria toxin without the massive inflammatory activation that is seen with lung injury induced by bacterial infections, viral infections, or lipopolysaccharide. The data convincingly demonstrate that there is an emergence of a highly proliferative lung endothelial subpopulation with specific molecular signatures that facilitate regeneration.

---

## [Decision Letter]

**Decision letter after peer review:**

Thank you for submitting your article "Single Cell Transcriptomic Atlas of Lung Microvascular Regeneration after Targeted Endothelial Cell Ablation" for consideration by *eLife*. Your article has been reviewed by 3 peer reviewers, including Jalees Rehman as the Reviewing Editor and Reviewer #1, and the evaluation has been overseen by Paul Noble as the Senior Editor. The following individuals involved in the review of your submission have agreed to reveal their identity: Vinicio de Jesus Perez (Reviewer #2); Wolfgang M. Kuebler (Reviewer #3).

Essential revisions:

1) Compare scRNA-seq on lung ECs with other recent lung injury endothelial scRNA-seq datasets to identify commonalities and differences in the signatures of clusters related to injury and proliferation responses.

2) Analyze the scRNA-seq changes in non-endothelial cells and identify potential cell-cell interactions between endothelial and non-endothelial cells

3) Perform a more in-depth analysis of changes in endothelial cell states using trajectory building and RNA velocity analyses as well as infer potential transcription factors driving the changes to assess whether there are additional factors beyond FoxM1 that are activated in the proliferative cluster

4) Characterize the apelin signaling pathway and its role in driving the shift of endothelial cells to stem-like states

5) Define the stem-like endothelial cells by isolation or in vitro induction, as well as the cues that result in their formation after the loss of adjacent endothelial cells post DT – is it the apoptosis of neighboring endothelial cells or is it the lack of signaling input from neighbors such as junctional cues, or a different mechanism.

*Reviewer #1 (Recommendations for the authors):*

The study could be strengthened by addressing the following points:

1. Clarify the "stem-like" nature of the cells. Is Procr just a marker of the stem-like cells endothelial cells or is it involved in the stem-like nature. "stem-like" really refers to multipotency – can these cells become non-endothelial cells? Or are they merely progenitors of endothelial cells and can only mature into endothelial cells? Providing experimental evidence for the "stem-like" nature by performing differentiation assays or replacing "stem-like" with a more appropriate term may be helpful.

2. Mechanistic studies: Apelin and Procr are repeatedly mentioned as two key markers of the new transition process that the endothelial cells undergo but the studies remain mostly descriptive except for the apelin inhibitor study which only measures mortality but does not track actual endothelial regeneration. Overexpression of apelin or Procr in vivo using lung endothelial-specific gene delivery (or shRNA delivery to block) and measurement of lung EC regeneration would help define their mechanistic roles. Alternatively, one could consider ex vivo studies on lung ECs to demonstrate potential mechanistic roles for the gCap transition process.

3. Bioinformatic analysis of ECs: One key claim is the transition of gCap ECs into the proliferative EC subpopulation. How does this observation relate to the emergence of proliferative lung EC populations following lung injury described by others in the field? Can one model the trajectory of the transient "stem-like" and proliferative EC populations using Monocle3 or other trajectory-building algorithms?

4. FoxM1: An unbiased analysis of the potential transcription factors involved in cell transition and proliferation activation would be helpful. This could be performed by algorithms that infer transcription factor activities in single-cell data (so that one does not have to rely on mRNA levels of transcription factors). it would be very compelling if FoxM1 was one of the top inferred transcription factors by such an unbiased analysis.

5. Non-endothelial cells: The authors have very valuable non-endothelial single-cell data which shows changes in cell numbers but there is no analysis of the transcriptome change in those cells. if the authors can show how endothelial ablation itself affects the gene expression in alveolar macs, epithelial cells, and other immune cells, then it would have broad implications for our understanding of how endothelial cells regulate lung homeostasis.

*Reviewer #2 (Recommendations for the authors):*

The manuscript titled "Single Cell Transcriptomic Atlas of Lung Microvascular Regeneration after Targeted Endothelial Cell Ablation" by Soares Godoy et al. underlines the importance of single-cell RNA seq in the discovery of mechanisms in lung microvascular repair. Overall, the study is well-designed, and the experiments are concise and relevant. However, there are certain areas where further explanation is required to greatly improve the quality of the manuscript and make it suitable for publication.

1. One of the major points referenced in the introduction is the role of ALI/ARDS in Sars-Cov-2 infection. I wonder if it would be possible to compare the datasets obtained from the DT-mouse study and the recently published scRNA-seq analysis of COVID lungs (doi.org/10.1038/s41586-021-03569-1) to identify whether there are similar cellular/genetic/molecular changes. This will greatly strengthen the implications of this animal study's findings and help mitigate the weaknesses associated with the model.

2. Could the authors obtain access to lung tissue from ALI/ARDS patients (+/- COVID) and stain for aCap/gCap and apelin molecules? It would be interesting to see whether these cell types exhibit patterns similar to animal ones.

3. The scRNA-seq dataset is incredibly rich and will be a major resource for the community. While the focus is understandably on ECs, I think attention should be given to cell-cell interactions across the lung. Could the authors carry out ligand-receptor analysis to further emphasize how other cell types react to the injury? Please see doi.org/10.1038/s41576-020-00292-x for details regarding the methodology that can be used to carry out this analysis.

*Reviewer #3 (Recommendations for the authors):*

1. The authors propose apelin-apelin-receptor signaling between "stem-like cells" (Cluster 1, zone 2) and "progenitor-like cells" (cluster 1, zone 3, and cluster 7) as a driver of endothelial regeneration, yet these cells seem to exist at different time points (in fact, the authors propose that the latter emerge from the former). So, how can cell A signal to cell B if at the same time cell A already transitions into cell B by way of this signaling?

2. For cluster 1, I am surprised that the authors did not use RNA velocity analyses to validate the time-dependent transition of cells across the different zones in this cluster. Such analyses would clearly strengthen the conclusion that this is a single cell type that is undergoing different cell states during this process.

3. What is the signal that activates these "stem-like cells" and makes them transition from basal gCap into cluster 1, zone 2 cells? Is it the apoptosis of adjacent endothelial cells, or the concomitant inflammation? While it may be difficult to answer this question, some discussion may be warranted as identification of such a signal would allow replicating – e.g., for in-depth studies but also for therapeutic purposes as suggested by the authors – this transition in vitro.

4. The mortality data comparing young vs. old mice and the effects of the apelin receptor antibody are interesting, but more insight at the cellular level would help to further validate (or refute) the proposed concept. Specifically, one may ask whether age or apelin receptor inhibition can prevent the formation of Cluster 1 zone 3 cells at Day 5 post-DT? I realize that, unfortunately, such an experiment is prevented by the fact that most of these mice die on day 4, and the residual mice may present a survival bias. But how do the authors reconcile the fact that these mice die at D4, while the actual apelin-receptor positive population (cluster 1 zone 3) which supposedly is targeted by the apelin receptor antibody only emerges on D5?

5. In Figure 1C, it would be helpful to show that the apoptotic cells are indeed endothelial cells by counterstaining with CD144, CD31, etc. rather than DAPI alone.

6. In their section on Speculations and Ideas, the authors propose isolating stem-like and progenitor-type gCap cells at different time points after DT treatment. Indeed, such isolation would go a long way to validate the proposed non-proliferative nature of the former vs. the high-proliferative nature of the letter by actual functional assays rather than transcriptomic profiles alone. This opportunity should at least be discussed.

---

## [Author Response]

Essential revisions:1) Compare scRNA-seq on lung ECs with other recent lung injury endothelial scRNA-seq datasets to identify commonalities and differences in the signatures of clusters related to injury and proliferation responses.

As described in detail in response to Reviewers 2 (Comment 1), we used a publicly available database from a recently published model of LPS induced ALI (DOI: 10.1172/jci.insight.158079) to perform a similar temporal analysis to compare the evolution of EC populations and gene signatures after LPS injury to our DT-induced ablation model. This revealed remarkably comparable cell populations essentially replicating the same profiles described in our report, notably the early emergence of an apelin-EPCR positive gCap EC cluster at early time points followed by the transition to a proliferative FoxM1-postive population. These new data are presented in a new Figure 6, Figure Supplement 3 and confirm that the regenerative endothelial populations that we have identified post EC ablation are indeed relevant to a commonly used model of ALI.

2) Analyze the scRNA-seq changes in non-endothelial cells and identify potential cell-cell interactions between endothelial and non-endothelial cells

As requested, we have performed further analyses on the major non-endothelial populations and developed a new high-resolution global UMAP now presented in a new Figure 3. We can now resolve 35 separate cell clusters, 26 of which represent non-endothelial cells. We have also performed an analysis of cell-cell interactions using NicheNet (doi.org/10.1038/s41592-019-0667-5), the results of which are presented in a new Figure 9. At Day 3, only one cluster – transitional cluster 1 – met our criteria for a ‘receiver’ population (>50 DEGs) (panels A and B) with the top 10 ligands (signals) originating from 21 clusters encompassing all global cell populations (stromal, myeloid, lymphoid, epithelial and endothelial). At Day 5, there was a substantial increase in cell-cell interactions, with 5 endothelial receiver populations responding to signals from 31 sender populations as shown in panels C and D. This is described in more detail in the response to the Reviewers.

3) Perform a more in-depth analysis of changes in endothelial cell states using trajectory building and RNA velocity analyses as well as infer potential transcription factors driving the changes to assess whether there are additional factors beyond FoxM1 that are activated in the proliferative cluster

As described more fully in our responses to Reviewers 1 (Comment 3 and 4) and 3 (Comment 2), we have now used the Python package scVelo to evaluate the RNA velocity and trajectory analysis interactions (new Figure 5). In brief, these new analyses validate the transition of EC regenerative populations within Zones 2 and 3 of cluster 1 ultimately repopulating depleted capillary, as well as arterial and venous EC populations. This is described in more detail in the responses to Reviewers #1 and #3 below.

In addition, we used decoupleR to explore the activity of relevant transcription factors based on prior knowledge of transcription factors-gene set. This analysis confirmed that FoxM1 was among the most prominent transcription factors (Figure 6 —figure supplement 2). See response to Reviewer #1 for a more detailed description.

4) Characterize the apelin signaling pathway and its role in driving the shift of endothelial cells to stem-like states

While we provide in vivo evidence suggesting a functional role for the Apln-Aplnr pathway in microvascular repair, it should be recognized that we used apelin expression mainly as a marker to identify the regenerative EC population. Even if mechanistic studies did not ultimately support an important functional role, we would argue that this would not detract from the novelty and utility of our data. Nonetheless activation of the major apelin signaling pathways was seen in Zones 2 and 3 of Cluster 1 at Days 3 and 5, respectively (see Figure 7), which is consistent with a functional role for this pathway. We also provide a chord diagram to better illustrate the genes that are activated or inhibited in Cluster 1 during the emergence of this EC population (i.e., Day 3 vs. Day 0; Figure 8, Figure Supplement 2). This analysis shows that three major apelin signaling pathways were activated during this transition (PI3/Akt, mTor, Elf4/p70S6K). However, we fully acknowledge that this is largely observational, and agree that mechanistic studies would be needed to establish whether there is indeed a causal role for apelin signaling in mediating the shift in endothelial stem-like states. We are undertaking these studies using isolated EPCR-positive cells and EPCR mutant transgenic mice, but expect that this will require a substantial of time and resources to complete and respectively suggest that this is beyond the scope of the present manuscript.

5) Define the stem-like endothelial cells by isolation or in vitro induction, as well as the cues that result in their formation after the loss of adjacent endothelial cells post DT – is it the apoptosis of neighboring endothelial cells or is it the lack of signaling input from neighbors such as junctional cues, or a different mechanism.

We agree that the isolation of the apelin/EPCR-positive stem-like ECs and defining the cues that result in their formation represent important and logical next steps to follow up on the novel findings contained in the present manuscript. We are making progress in isolating and characterizing these cells; however, this will take considerable time and resources to successfully complete. Therefore, we believe that this work should form the basis of separate follow-on publication. As Reviewer 3 notes, “while it may be difficult to answer this question, some discussion may be warranted as identification of such a signal would allow replicating – e.g., for in-depth studies but also for therapeutic purposes as suggested by the authors – this transition in vitro”. Therefore, we have expanded the discussion to better outline how this might be accomplished (page 17, lines 510-522).

Reviewer #1 (Recommendations for the authors):The study could be strengthened by addressing the following points:1. Clarify the "stem-like" nature of the cells. Is Procr just a marker of the stem-like cells endothelial cells or is it involved in the stem-like nature. "stem-like" really refers to multipotency – can these cells become non-endothelial cells? Or are they merely progenitors of endothelial cells and can only mature into endothelial cells? Providing experimental evidence for the "stem-like" nature by performing differentiation assays or replacing "stem-like" with a more appropriate term may be helpful.

We fully acknowledge that we have not definitively demonstrated that this population fulfils all the criteria of a true stem cell. However, we used the term, ‘stem-like’ because they do express markers that have previously used to identify resident vascular endothelial stem cells (VESCs) (doi: 10.1038/cr.2016.85) and hematopoietic stem cells (DOI 10.1182/blood-2005-06-2249). As well, we have shown that they can regenerate all lung EC subtypes, including arterial, venous, gCap and highly specialized aCap ECs (or aerocytes), which are critical for re-establishing the air-blood barrier. As such, the exhibit some features of a potential endothelial stem cell population. While we agree that to call them stem cells would require in vitro characterization of their self-renewal and differentiation potential (which is underway), we respectfully submit that the use of the term ‘stem-like’ is reasonable and justified since it clearly acknowledges to the reader this limitation while highlighting their potential regenerative nature.

2. Mechanistic studies: Apelin and Procr are repeatedly mentioned as two key markers of the new transition process that the endothelial cells undergo but the studies remain mostly descriptive except for the apelin inhibitor study which only measures mortality but does not track actual endothelial regeneration. Overexpression of apelin or Procr in vivo using lung endothelial-specific gene delivery (or shRNA delivery to block) and measurement of lung EC regeneration would help define their mechanistic roles. Alternatively, one could consider ex vivo studies on lung ECs to demonstrate potential mechanistic roles for the gCap transition process.

In the present manuscript these two gene products were used mainly as markers to identify the transient regenerative EC population, although, as the Reviewer notes, the effect of inhibition of the apelin is consistent with a functional role for this pathway. While we are pursuing both in vivo and in vitro strategies to better define the functional importance of the apelin and EPCR pathways in microvascular repair, this represents a substantial additional body of new experimental work that we would suggest is beyond the scope of this first report. Nonetheless, even if the results of these mechanistic studies ultimately do not support a functional role in microvascular repair, we would argue that this would not negate the utility of these markers to identify key regenerative cell populations nor detract from the novelty and importance of the data described in the present manuscript.

3. Bioinformatic analysis of ECs: One key claim is the transition of gCap ECs into the proliferative EC subpopulation. How does this observation relate to the emergence of proliferative lung EC populations following lung injury described by others in the field?

The transition to a highly proliferative endothelial population that expresses FoxM1 (among other pro-proliferative TFs) is very consistent with work from others that has implicated FoxM1 lung endothelial repair and resolution of ALI in polymicrobial and LPS models (doi:10.1002/stem.1690; DOI: 10.1371/journal.pone.0050094; doi.org/10.1101/2021.04.29.442061) as well as in models of bronchopulmonary dysplasia (DOI: 10.1164/rccm.201906-1232OC).

Can one model the trajectory of the transient "stem-like" and proliferative EC populations using Monocle3 or other trajectory-building algorithms?

Based on the Reviewer’s suggestions, we have used the Python package scVelo to evaluate the RNA velocity and trajectory analysis within our EC clusters. This dynamical model showed a progressive shift of ECs through Zones 2 and 3 of transitional Cluster 1 with strong vectors towards depleted EC populations, including gCap ECs of Cluster 0, arterial and venous EC populations (new Figure 5A). Notably, Zone 2 cells exhibited evidence of DNA synthesis (panel B), whereas cells in Zone 3, and the proliferative Cluster 7, showed increased G2M score, consistent with EC proliferation. Moreover, we identified the top 15 genes involved in driving these trajectories (panel C), with 8 genes implicated in the transition from cluster 1 to cluster 0, 5 genes related to the transition from Cluster 1 to aCap ECs/aerocytes (Cluster 2) and 2 genes for Cluster 1 to venous ECs. Finally, an inferred latent time plot of the 300 top driver genes revealed the same temporal progression of gene expression as observed with the independent timepoint analysis (Panel D). Therefore, despite the fact that this velocity analysis infers a timeline based only on genomic cues (i.e., proportion of spliced RNA), and is agnostic to the actual timing of the cell harvesting in the serial transcriptomic dataset, it approximates remarkably well the temporal shifts in EC populations that were demonstrated by serial scRNA-seq at the defined timepoints.

4. FoxM1: An unbiased analysis of the potential transcription factors involved in cell transition and proliferation activation would be helpful. This could be performed by algorithms that infer transcription factor activities in single-cell data (so that one does not have to rely on mRNA levels of transcription factors). it would be very compelling if FoxM1 was one of the top inferred transcription factors by such an unbiased analysis.

We thank the Reviewer for this comment and agree that, due to their low expression levels, many transcription factors (TFs) may have been overlooked based on their mRNA levels alone. We used the decoupleR software package to infer TF activity based on known TF-gene set interactions. This revealed 25 TFs that are significantly up or downregulated in our dataset, with very distinct profiles in the different ECs populations (Figure 6, Supplement Figure 2). Not surprisingly, the greatest increase in TF activity was seen in the proliferative Cluster 7 with FoxM1 indeed being among the top candidates. Other TFs known to be involved in cell cycle regulation were also identified, including three members of the E2f family, Myc and Tfdp1. As well, there were some surprising findings such as strong Nanog and Sox 10 activity seen only in aCap ECs. These TFs are known to play important roles in stem cell and developmental biology. In Cluster 3, which is characterized by high expression of early response genes such as c-Fos, we saw strong activity of the Ets family member, Elk4, which binds to the serum response element in the promoter of the c-Fos. Of note, the same four TFs found in gCap-ERG ECs were also seen in the arterial (and less so venous) cluster, but not in other capillary ECs, further confirming the transcriptional heterogeneity of capillary EC populations.

5. Non-endothelial cells: The authors have very valuable non-endothelial single-cell data which shows changes in cell numbers but there is no analysis of the transcriptome change in those cells. if the authors can show how endothelial ablation itself affects the gene expression in alveolar macs, epithelial cells, and other immune cells, then it would have broad implications for our understanding of how endothelial cells regulate lung homeostasis.

We appreciate the reviewer’s comments and agree that the remarkable changes in non-endothelial populations in response to EC ablation has broad implications. We have performed a subset analysis on non-endothelial populations with higher resolution nomenclature and breakdown of significant temporal changes after EC injury and during lung repair, which is now presented in new Figure 3. While the greatest transcriptomic changes were seen in EC populations, a number of non-EC populations were in the top 10 most affected including alveolar and interstitial macrophages, pericytes and type 2 alveolar epithelial cells. We also assessed cell-cell interactions between non-endothelial populations and ECs in a new Figure 9 as described in detail in response to Reviewer 2.

Reviewer #2 (Recommendations for the authors):The manuscript titled "Single Cell Transcriptomic Atlas of Lung Microvascular Regeneration after Targeted Endothelial Cell Ablation" by Soares Godoy et al. underlines the importance of single-cell RNA seq in the discovery of mechanisms in lung microvascular repair. Overall, the study is well-designed, and the experiments are concise and relevant. However, there are certain areas where further explanation is required to greatly improve the quality of the manuscript and make it suitable for publication.1. One of the major points referenced in the introduction is the role of ALI/ARDS in Sars-Cov-2 infection. I wonder if it would be possible to compare the datasets obtained from the DT-mouse study and the recently published scRNA-seq analysis of COVID lungs (doi.org/10.1038/s41586-021-03569-1) to identify whether there are similar cellular/genetic/molecular changes. This will greatly strengthen the implications of this animal study's findings and help mitigate the weaknesses associated with the model.

We thank the Reviewer for their excellent suggestion to compare the dataset from the DT mouse model with other recently published datasets. The Reviewer specifically suggested that we compare our dataset with a recently published human study in COVID ARDS. However, EC populations were grossly underrepresented in this dataset which would make identification of our transient EC populations challenging. Moreover, this transcriptomic data was only obtained from autopsy material and thus represents only the very late stages of disease. Therefore, we choose to use a dataset from a recently published model of LPS induced ALI (DOI: 10.1172/jci.insight.158079) to assess the relevance our model to commonly used preclinical models of ALI. In this report, the authors explored transcriptional changes in EC populations at multiple time points after LPS-induced lung injury and during repair, but separate analyses were performed at each timepoint. Instead, we analyzed this dataset in aggregate over time in the similar way that we analyzed our dataset in order to make a direct comparison. Although the time course was slightly different, we could identify very similar transient gCap EC populations occurring in the same sequence as in our dataset. Specifically, we observed a population of Cd93+ gCap ECs expressing both Apln and Procr, appearing as early as 6h post LPS, followed by a cluster of FoxM1+, proliferating ECs appearing at 3 days (Figure 6, Figure supplement 3). The earlier appearance of the regenerative EC populations in this model may relate to a more rapid action of LPS compared with DT-induced injury. Nonetheless, there was a very similar temporal sequence from Apln/Procr positive, gCap ECs transitioning to highly proliferative FoxM1-positive progenitors which strongly supports the relevance of our findings more broadly in other ALI models.

2. Could the authors obtain access to lung tissue from ALI/ARDS patients (+/- COVID) and stain for aCap/gCap and apelin molecules? It would be interesting to see whether these cell types exhibit patterns similar to animal ones.

We agree that it would be interesting to try to identify similar regenerative EC populations in sections of human lung from ARDS patients. Unfortunately, we have not yet been able to suitable human specimens for this purpose.

3. The scRNA-seq dataset is incredibly rich and will be a major resource for the community. While the focus is understandably on ECs, I think attention should be given to cell-cell interactions across the lung. Could the authors carry out ligand-receptor analysis to further emphasize how other cell types react to the injury? Please see doi.org/10.1038/s41576-020-00292-x for details regarding the methodology that can be used to carry out this analysis.

We thank the Reviewer for this comment and for the reference. As described in our response to Reviewer 1, we have performed a higher resolution analysis of the non-EC populations and have identified a number of interesting cell populations that have clear transcriptomic changes during injury-repair (New Figure 3). As requested, we have performed a ligand-receptor analysis using NicheNet focusing on endothelial receiver populations during the critical period for endothelial regeneration and these results are presented in the new Figure 9. At Day 3, only the transitional ECs of Cluster 1 (Zone 2) met the criteria for an endothelial ‘receiver’ population (i.e., >50 DEGs vs. Day 0) (Figure 9A and B) largely representing apelin/Procr co-expressing gCap ECs. The top 10 ligands originated from 21 individual clusters representing all major cell populations (stromal, myeloid, lymphoid, epithelial and endothelial). Interestingly, the top four predicted ligands based of Pearson correlation coefficients, Occludin (Ocln), FAT Atypical Cadherin 1 (Fat1), Plexin B2 (Plxnb2) and Ephrin B1 (Efnb1), mediate signaling by direct cell-cell contact and play major roles in developmental biology, including regulation of proliferation, migration and differentiation. Ocln and Efnb1 were associated with interactions between other EC subtypes, whereas Plxnb2 was exclusively involved in cell-cell signaling between ECs and non-EC clusters including pericytes, monocyte/macrophages and epithelial cells. In contrast, Fat1 was restricted to interactions between EC and pericytes and smooth muscle cells. At Day 5, there was a substantially greater diversity in cell-cell interactions, with five endothelial receiver populations responding to ligands from 31 cell clusters representing all cell populations (Figure 9C and D). Again, transitional gCap ECs (Cluster 1, Zone 3) exhibited the strongest predictions for the top ligands which, with the exception of Fat1, remained the same as for Day 3. The other endothelial receiver clusters showed considerable differences in their top ranked ligands, likely reflecting the distinct pathways involved in the differentiation of Cluster 1 ECs into these specialized subpopulations during the microvascular repair.

Reviewer #3 (Recommendations for the authors):1. The authors propose apelin-apelin-receptor signaling between "stem-like cells" (Cluster 1, zone 2) and "progenitor-like cells" (cluster 1, zone 3, and cluster 7) as a driver of endothelial regeneration, yet these cells seem to exist at different time points (in fact, the authors propose that the latter emerge from the former). So, how can cell A signal to cell B if at the same time cell A already transitions into cell B by way of this signaling?

The temporal dissociation of expression of apelin and its receptor is in large part artifactual due to the fact that we only measured expression at discreet timepoints that were 2 days apart. While a small number of cells can still be seen expressing apelin at day 5, it is very likely that the overlap in expression of apelin and its receptor was more substantial in between these timepoints.

2. For cluster 1, I am surprised that the authors did not use RNA velocity analyses to validate the time-dependent transition of cells across the different zones in this cluster. Such analyses would clearly strengthen the conclusion that this is a single cell type that is undergoing different cell states during this process.

Based on the Reviewer’s suggestions, we have used the Python package scVelo to evaluate the RNA velocity and trajectory analysis within our EC clusters. This dynamical model showed a progressive shift of EC populations within transitional cluster 1 through zones 2 and 3 with strong vectors towards depleted populations (new Figure 5), including gCap ECs of Cluster 0, and arterial and venous EC populations. Notably, zone 2 cells uniquely exhibited evidence of DNA synthesis based on the S score, whereas cells in zone 3, and the proliferative cluster 7, showed increased G2M score, indicative of proliferation (panel B). We also identified the top 15 driver genes involved in specific transitions (panel C), with 8 genes implicated in the transition from cluster 1 to cluster 0, 5 genes involved in transition from Cluster 1 to aCap ECs (Cluster 2) and 2 genes for Cluster 1 to venous ECs. Finally, an inferred latent time plot of the 300 top driver genes replicated the same temporal progression of dynamic gene expression that was seen by serial transcriptomic analysis beginning with Cluster 1 and progressing through to Cluster 0 and finally Cluster 2 at the end of the pseudotime cascade (Panel D). Therefore, despite the fact that this velocity analysis only infers a timeline based on transcriptional cues (i.e., proportion of spliced to unspliced RNA), and is agnostic to the actual timing of the cell harvesting in the serial transcriptomic dataset, it approximates remarkably well the temporal shifts in EC populations that were demonstrated by serial scRNA-seq at defined timepoints.

3. What is the signal that activates these "stem-like cells" and makes them transition from basal gCap into cluster 1, zone 2 cells? Is it the apoptosis of adjacent endothelial cells, or the concomitant inflammation? While it may be difficult to answer this question, some discussion may be warranted as identification of such a signal would allow replicating – e.g., for in-depth studies but also for therapeutic purposes as suggested by the authors – this transition in vitro.

This is an important question; however, as recognized by the Reviewer, it will not be easy address. It is unlikely that simply overexpressing apelin, Procr or both in cultured ECs will be sufficient to induce a stem-like phenotype. Indeed, Procr/EPCR overexpression in hematopoietic stem cells failed to replicate the higher proliferation and engraftment potential of HSCs that spontaneously express EPCR (DOI 10.1182/blood-2016-11-750729). We are in the process of optimizing the isolation and culture of these cells by EC selection based on EPCR expression, which will allow a comprehensive exploration of the cues leading to the generation of these cells after endothelial injury in vitro using transcriptomic, proteomic and high-throughput technologies. While this is a logical follow-on based on our novel findings, we would respectfully suggest that it represents new work beyond the scope of present manuscript. However, as the Reviewer suggests, we have revised the discussion to highlight how this question might be addressed in these future studies (Page 17, lines 510-522).

4. The mortality data comparing young vs. old mice and the effects of the apelin receptor antibody are interesting, but more insight at the cellular level would help to further validate (or refute) the proposed concept. Specifically, one may ask whether age or apelin receptor inhibition can prevent the formation of Cluster 1 zone 3 cells at Day 5 post-DT? I realize that, unfortunately, such an experiment is prevented by the fact that most of these mice die on day 4, and the residual mice may present a survival bias. But how do the authors reconcile the fact that these mice die at D4, while the actual apelin-receptor positive population (cluster 1 zone 3) which supposedly is targeted by the apelin receptor antibody only emerges on D5?

Again, as stated in our response to Comment 1, the relative lack of overlap in cells expressing the ligand and receptor is likely related to the fact we performed scRNA-seq only at 3 and 5 days. In fact, even at 5 days there are some cells that continue to express apelin. No doubt there would be a much greater overlap in cells expressing apelin or its receptor at 4 days.

5. In Figure 1C, it would be helpful to show that the apoptotic cells are indeed endothelial cells by counterstaining with CD144, CD31, etc. rather than DAPI alone.

We did try to show the apoptotic cells were endothelial cells using CD144 as a counterstain but, due to significant differences in the antigen retrieval requirements for the optimized antibodies, we found these antibodies were incompatible. However, we provide the Reviewer with a two-gene UMAP of the EC subset showing the co-expression of caspase 3 and cadherin 5 expression in the lung ECs, which clearly shows co-expression, Author response image 1.

**Author response image 1. sa2fig1:** 

6. In their section on Speculations and Ideas, the authors propose isolating stem-like and progenitor-type gCap cells at different time points after DT treatment. Indeed, such isolation would go a long way to validate the proposed non-proliferative nature of the former vs. the high-proliferative nature of the letter by actual functional assays rather than transcriptomic profiles alone. This opportunity should at least be discussed.

As suggested by the Reviewer, we now discuss the opportunity of isolation of stem-like gCap ECs to better characterize their regenerative potential and identify the cues that result in their development, which will be necessary to translate this work into potential therapeutic application (page 17, lines 510-522).